



# Hydrogeological responses to the 2016 Gyeongju earthquakes,
# Korea
Jaeyeon Kim[1], Jungjin Lee[1], Marco Petitta[2], Heejung Kim[1], Dugin Kaown[1], In-Woo Park[1],
Sanghoon Lee[1] and Kang-Kun Lee[1]*
[1] School of Earth and Environmental Sciences, Seoul National University, Seoul 08826, Republic of Korea.
[2] Department of Earth Sciences, Sapienza University of Rome, P.le A. Moro 5, 00185 Rome, Italy.
*Correspondence to*: Kang-Kun Lee[1]* (kklee@snu.ac.kr)




**Abstract**

9        The September 12, 2016 Gyeongju earthquakes (M5.1 and M5.8) had significant effects on

groundwater systems along the Yangsan Fault System composed of NNE-trending, right-
lateral strike-slip faults in Korea. Hydrological changes induced by the earthquakes are
important because no surface ruptures have been reported and few earthquakes usually occur
in Korea. The main objective of this research was to propose a conceptual model interpreting
the possible mechanisms of groundwater response to the earthquakes based on anomalous
hydrogeochemical data including isotope (radon, strontium) concentrations with bedrock
characteristics. To analyze the hydraulic changes resulting from the earthquakes, annual
monitoring data of groundwater level, temperature, and electrical conductivity and collected
data of hydrochemical parameters, radon-222, and strontium isotopes were collected during
January 2017. Groundwater level anomalies could be attributed to the movement of the
epicentral strike-slip fault. Radon concentration data showed the potential of groundwater
mixing processes. Strontium anomalies could be related to the lithology and stratigraphy of
the bedrock, reflecting the effect of water–rock interaction. Using a Self-Organizing Map
(SOM) statistical analysis, associations of hydro-geochemical characteristics among
groundwater wells were interpreted. By combining the grouped results of the SOM with
lithostratigraphic unit data, 21 groundwater wells were classified into four groups, each
corresponding to different hydrogeological behaviors. A new comprehensive conceptual
model was developed to explain possible mechanisms for the hydrological and geochemical



responses in each group, which have been respectively identified as water–rock interaction,
mixing of shallow and deep aquifers via sea water intrusion, bedrock fracture opening related
to strike-slip fault movement, and no response.



## 1. Introduction

Earthquakes have a great influence on underground hydrology, such as water table changes and groundwater chemistry anomalies. Typically, most studies have focused on earthquake forecasting, i.e. changes prior to earthquakes. There have been few studies that discuss the responses of groundwater systems following earthquakes (Adinolfi Falcone et al., 2012;Amoruso et al., 2011;Barberio et al., 2017;Claesson et al., 2007;Ekemen Keskin, 2010;Galassi et al., 2014;Lee et al., 2013;Matsumoto et al., 2003;Petitta et al., 2018;Wang and Manga, 2010;Wang et al., 2012;Yechieli and Bein, 2002). Seismicity can cause abrupt changes or have long-term effects on the environment, particularly, groundwater systems. Seismic waves, for example, are known to cause changes in water level, temperature, and geochemistry (Matsumoto et al., 2003;Roeloffs et al., 2003;Roeloffs, 1998;Shi et al., 2015;Adinolfi Falcone et al., 2012;Wang et al., 2012). Hydrological responses to seismicity depend on several factors such as the earthquake magnitude, distance from the epicenter, the chemical and physical properties of the water, geological structures, permeability, and the pore pressure of rocks (Ekemen Keskin, 2010;Hartmann and Levy, 2005;Petitta et al., 2018). For example, Ekemen Keskin (2010) stated that the observed changes in aquifers could be explained using a dilatancy-fluid model; the response to earthquakes could be attributed to the changes in the water mixing ratio because of aquifer permeability, pore pressure, and flow path. Moreover, locally heterogeneous responses of groundwater have been observed and associated with the dominant lithology and mineralogy of bedrocks (Frape et al., 1984;Shand



et al., 2009;Kim et al., 1996), local degree of deformation (Fitz-Diaz et al., 2011), or fracture
networks allowing groundwater flow (Gray et al., 1991). By using hydraulic properties, some
studies also have proposed some conceptual models for describing the aquifer responses to
earthquakes (Adinolfi Falcone et al., 2012;Amoruso and Crescentini, 2010;Amoruso et al.,
2011;Barberio et al., 2017;Manga, 2001;Roeloffs, 1998;Tokunaga, 1999).
On September 12, 2016, two earthquakes ($M_L$ = 5.1 and $M_L$ = 5.8, respectively) occurred in
Gyeongju, in the southeastern part of the Korean Peninsula (Korea Meteorological
Administration). The $M_L$ of the mainshock was recorded as the largest in Korea since seismic
monitoring started in Korea in 1978. The source mechanism of the Gyeongju earthquakes
displayed strike-slip movement of a branch of the Yangsan Fault (YSF) passing through the
Gyeongju area (Kim et al., 2017b;Kim et al., 2016). The occurrence of the Gyeongju
earthquakes was shocking to people, as Korea has predominantly been recognized as
seismically stable. Actually, the earthquake catalog over the past 2,000 years shows that
historically damaging earthquakes in Korea have mainly occurred in the southeastern part of
the peninsula, particularly in the Gyeongju area near the YSF (Lee and Jin, 1991;Lee and Na,
1983). Slip analysis and earthquake focal mechanism solutions have interpreted that the YSF
is under a regional compressional stress field which might be a result of the continental
collision of the Pacific, Eurasian, and Indian plates (Jiang et al., 2016;Park et al., 2007;Park
et al., 2006;Zoback, 1992). The occurrence of the Gyeongju earthquakes provides an
opportunity for a more detailed study of the YSF and post-earthquake changes including



groundwater responses, because some local hydrologic effects related to the lithology were
observed following the Gyeongju earthquakes.
The earthquake-related indicators in hydrogeology generally include (i) groundwater level,
(ii) temperature, (iii) hydrochemistry, and (iv) isotope concentrations. Groundwater level
monitoring has been broadly used to identify pre-, co-, and post- earthquake changes (Ben-
Zion and Aki, 1990;Brodsky, 2003;Manga et al., 2012;Roeloffs, 1998;Shi et al., 2015;Wang
and Manga, 2010). Seismic waves have been known to affect the groundwater level via
oscillations and permanent offsets. Temperature changes are commonly analyzed using
groundwater level data (Ekemen Keskin, 2010;Kitagawa and Koizumi, 2000). In this study,
groundwater chemistry, major elements, and some physical-chemical parameters (pH, EC,
and temperature) were monitored. Among the isotopes, oxygen, hydrogen, and radon-222
were analyzed to determine the effects of the earthquake on groundwater. Radon-222,
particularly, has been generally monitored as an earthquake precursor sampled in water or air
(Igarashi et al., 1995;King, 1978;Liu et al., 1984;Noguchi and Wakita, 1977;Roeloffs,
1999;Teng, 1980;Wakita et al., 1980). Radon-222 is a radioactive nuclide with a half-life of
approximately 3.8 days. It is produced from radium-226 in the natural radioactive decay
chain of uranium-228; thus, its concentration is proportional to the uranium concentration in
adjacent rocks. The transport of radon is influenced by fluid advection, diffusion, partition
between the liquid and gas phases, and radioactive decay. The radon concentration in
groundwater is dependent on the surface area of the rocks (Hoehn and Von Gunten,





1989;Torgersen et al., 1990). Because the surface area can be affected by earthquakes, the
radon concentration can increase or decrease. Radon-222 also shows significant anomalies at
fault zones prior to earthquakes (Ghosh et al., 2009;Walia et al., 2009;Wang and Fialko,
2015). However, few studies have delineated the response of radon concentration to
earthquakes. Strontium isotopes have been used in only a few earthquake-related papers.
These isotopes are useful tracers for groundwater origin and water–rock mixing processes
because $^{87}$Sr is the daughter product of the natural decay of radioactive $^{87}$Rb (half-life = 48.9
Ga). The $^{87}$Sr/$^{86}$Sr ratio differs according to the rock type in the bedrock of aquifers (Frape et
al., 1984;Frost and Toner, 2004;Négrel et al., 2004;Shand et al., 2009).   Thus, strontium
isotopes in groundwater can also reveal significant post-earthquake anomalies at fault zones
according to bedrock type.
The main objective of this study was to identify hydrogeochemical changes related to the
Gyeongju earthquake and then suggest a conceptual model of the response of the
groundwater systems to the earthquake using the grouping results. In accordance with this
objective, major research results were achieved via (i) performing a correlation and cluster
analysis of hydrochemical parameters with geological characteristics using the SOM
approach; (ii) analyzing pre-, co-, and post-seismic changes in groundwater level,
temperature, and EC; and (iii) interpreting the results of isotopes (radon and strontium)
sampled following the earthquake based on the grouping. These results could help to provide
the possible mechanisms of groundwater changes induced by the earthquake. The overview



of this research is shown in Fig. 1.

**2.  Study area**
The Gyeongju earthquake sequence started with a foreshock ($M_L$ = 5.1) at 10:44:32 UTC,
on September 12, 2016, and the mainshock ($M_L$ = 5.8) occurred at 11:32:55 UTC (Korea
Meteorological Administration). During the first 10 days following the mainshock, more than
120 earthquakes of $M_L \geq 2.0$ were recorded in the epicentral region. The earthquakes,
including the mainshock and strong aftershocks ($M_L \geq 3.5$) are listed in Table 1.
The Korean Peninsula is composed of three major Precambrian massifs: the Nangrim,
Gyeonggi, and Youngnam from north to the south. The Gyeongsang Basin is the northern
part of the Youngnam massif and the Yangsan Fault System has developed in the eastern part
of the Gyeongsang Basin. The Yangsan Fault System is a group of NNE-trending major
strike-slip faults. The Gyeongju earthquake and its abundant aftershocks occurred near the
YSF (Fig. 2a), which has a linear expression for approximately 200 km, and is the longest
major fault of the Yangsan Fault System (Kyung and Lee, 2006). The displacement of the
fault is approximately 21–35 km depending on the location, and the arrangement of the
granitic rocks in this area indicates Cenozoic dextral strike-slip of 21.3 km in the N 20° E
direction along the YSF line (Hwang et al., 2004;Hwang et al., 2007).



Since Lee and Na (1983) first suggested that a Quaternary reactivation of the Yangsan Fault
System could be possible, a number of seismic, geological, and geophysical studies have
proved its seismic activation (Kyung and Lee, 1999;Kyung and Lee, 2006;Lee and Jin, 1991).
The Gyeongju area has been subject to most of the large historical earthquakes that have
occurred in Korea. Initial movement on the YSF was recorded to have occurred before 45
Ma, based on radiometric dating of volcanic rocks (Chang et al., 1990). Age dating using the
accelerator mass spectrometry (AMS) method indicates late Quaternary movement of the
YSF between 2,400 and 2,000 yrs BP and an average vertical slip rate of approximately 0.04–
0.05 mm/yr (Kyung and Chang, 2001). Recent measurement of the vertical slip rate of YSF
reported less than 0.1 mm/yr on average (0.02–0.07 mm/yr in the southern part and 0.03–0.05
mm/yr in the northern part) (Kyung, 2003;Kyung and Lee, 2006), indicating that the YSF has
been seismically active.
Paleo-stress analyses have noted that the stress regime of the YSF has changed more than
three times (Kang and Ryoo, 2009;Kim et al., 1996), and during the Quaternary the ENE-
WSW maximum compression is in agreement with the first-order stress field in east Asia
(Chang et al., 2010;Heidbach et al., 2010;Zoback, 1992). Trench analysis of the Yugye Fault,
the youngest Quaternary fault in the northern part of the YSF, also yielded a NW-SE or
WNW-ENE compressional local maximum principal stress (Kim and Jin, 2006). For the
Gyeongju event, also under this ENE-WNW compression, geophysical studies of the
aftershocks recognized the subsurface fault plane has a strike of NNE 25–30° and a dip of



65–74° with a depth ranging from 11 km to 16 km, The width of the distribution of event
locations is approximately 5 km in length, and was determined to be a branch of the YSF
(Hong et al., 2017;Kim et al., 2017a;Lee et al., 2018;Son et al., 2017).
Twelve wells are located near the YSF and the surrounding area within the Gyeongsang
Basin (Fig. 2b). The information for each well is shown in Table 2. The lithostratigraphic unit
indicates the characteristic of the bedrock aquifer wells (labeled as KW##-2). The
Gyeongsang Basin is mainly composed of Cretaceous and Tertiary non-marine sediments and
igneous rocks (Fig. 2b); Middle Cretaceous Hayang group sediments, Late Cretaceous
Yucheon group rocks, Early Miocene Yeonil group rocks, Middle Miocene Janggi group, and
Bulguksa group granitic rocks which intruded the Cretaceous rocks during the Late
Cretaceous to Early Tertiary (Chang, 1975, 1977, 1978;Chang et al., 1990). The lithology of
each stratigraphic unit as documented in detail by the Korea Institute of Geoscience and
Mineral Resources (KIGAM) can be briefly characterized as follows; the Hayang group is
mostly composed of clastic sedimentary rocks including shale, mudstone, and sandstone with
mafic or intermediate volcanic rocks. The Yucheon group consists of andesitic rocks and
quartz andesites including plagioclase phenocrysts. The Yeonil group and Janggi group
consist of Early and Middle Miocene sedimentary and volcanic rocks that are 'mainly
exposed in the eastern part of the Gyeongsang Basin. The Yeonil group basin consists of a
tuffaceous Tertiary sedimentary basin, and Miocene basal conglomeratic rocks, which consist
of light brown to light gray conglomerate and sandstone. The Janggi group rocks mainly





consist of basaltic tuff and andesitic tuff of Early Miocene age. The Bulguksa intrusive rocks
are mainly composed of biotite granites accompanying grano-diorite, tonalite, and alkali-
feldspar granites (Hwang et al., 2004). Based on the lithology and stratigraphy, this study
divided the bedrock aquifer well locations into four areas; (i) Hayang-group shale and
sandstone (KW 1, KW 2, KW 9-2, KW 10-2), (ii) Bulguksa-group biotite granite (KW 3, KW
5-2, KW 12-2), (iii) tuff and tuffaceous sedimentary rocks of Yeonil-group and Janggi group
(KW 4-2, KW 6-2, KW 7-2), and (iv) Cretaceous volcanic rocks mainly composed of
andesite (KW 8-2, KW 11-2).

**3.  Methods & materials**
**Water sampling and analysis method**
Continuous monitoring data for water level, temperature, and electrical conductivity (EC),
from the National Groundwater Monitoring Network (NGMN) of the Korea Water Resources
Corporation, were analyzed (http://www.gims.go.kr). These daily data, which correspond to
the average of the values measured every hour, were used for observing the responses before,
during, and after the earthquake. Precipitation data obtained from the Korea Meteorological
Administration were also analyzed with the water level variation (http://kma.go.kr).
Sampling of groundwater wells in alluvial and bedrock aquifers was conducted for three
days (January 16, 2017 to January 18, 2017) four months after the earthquake. A total of 22





water samples, including one of surface water, were collected in 2-L polyethylene bottles
using a Grundfos MP-1 pump. The samples were analyzed for hydrochemical parameters,
major ions, radon concentration, and strontium isotopes. The hydrochemical parameters
temperature, EC, dissolved oxygen (DO), total dissolved solids (TDS), pH, and salinity were
measured in the field using an YSI ProDSS digital sampling system (Xylem, USA). The
analysis of cations and anions (Na, K, Ca, Mg, Cl, $NO_3$, $SO_4$, and $HCO_3$). including strontium
isotopes, was completed using filtered water samples at the Korea Basic Science Institute
(KBSI). $^{87}Sr/^{86}Sr$ ratios were obtained using a Neptune Multicollector-Inductively Coupled
Plasma Mass Spectrometer (MC-ICP-MS; Thermo Finnigan, Germany) upgraded with a large
dry interface pump. Yields were approximately 100% and the matrix concentration did not
exceed 1% of the strontium concentrations. The total procedural blanks were negligible with
less than 1 ng of Sr. The $^{87}Sr/^{86}Sr$ ratios were normalized to $^{86}Sr/^{88}Sr = 0.1194$ (Faure, 1986),
and the mean $^{87}Sr/^{86}Sr$ ratio of NBS987 (U.S. National Bureau of Standards) was $0.710247 \pm$
$0.000017$ ($2\sigma$, n = 18). The radon concentrations in the groundwater samples were measured
using RTM1688-2 of SARAD. An air-bubbling 500-ml flask was filled with sampled water
and connected to the monitor for a closed air loop. The measurement was conducted at 15-
min intervals. The attained values were calibrated adjusting for the short half-life of radon.
The unit offers a high sensitivity of better than 3 cpm/(kBq/m³) obtained from a very small
internal volume of only 130 ml.
**Self-Organizing Map (SOM)**





211 Self-Organizing Map (SOM) analysis is a neural network organized on a low-dimensional

212 array of processing units (Kohonen, 1982). The SOM consists of two layers: the input layer

213 and the output layer (neurons layer). These two layers are interconnected via a weight vector.

214 The neurons in the output layer are connected to adjacent neurons by a neighborhood relation

215 dictating the structure of the topographic map. In this study, the layer of neurons was

216 arranged onto a two-dimensional grid. The SOM is an unsupervised learning algorithm

217 without prior information of classification. The learning algorithm procedure can be

218 described as follows: (i) determine the number of neurons, (ii) initialize the weight vectors

219 with small random values, (iii) choose the best-matching neurons or the best-matching unit

220 (BMU) that is the closest to the input vector, and (iv) update the best-matching neurons and

221 neighboring neurons. The results can be visualized using two different types of map: the

222 component planes and the U-matrix (Vesanto, 1999). The component plane representation

223 visualizes relative component values of the weight vectors, providing correlations between

224 components. The U-matrix, i.e. the unified distance matrix, enables clustering analysis using

225 the distance between the weight vectors and their neighborhood. The simulation was

226 completed using the SOM toolbox 2.0 for Matlab 5 (Vesanto and Alhoniemi, 2000).


228 **4. Results**

229 **4.1 Self-Organizing Map (SOM)**





There are few studies using SOM for groundwater quality data interpretation (Choi et al.,
2014;Hong and Rosen, 2001;Lischeid, 2008). However, we used the SOM analysis for
statistical analysis in the Gyeongju area because it can solve linear dimensionality reduction
problems without biases. The contribution map of the variables is shown in the component
map (Fig. 3). The dataset contained data regarding 16 variables (Na, K, Ca, Mg, Cl, $NO_3$, $SO_4$,
$HCO_3$, Sr, $^{87}Sr/^{86}Sr$, temperature, pH, DO, EC, TDS, and salinity) (Table 3). The values were
used in a normalization form, not raw data. The normalization 'var', a simple linear
transformation, was used with the variance of the variable to unity and its mean to zero. By
comparing component planes, the planes of Ca, $SO_4$, Sr, and $^{87}Sr/^{86}Sr$ have similar
distributions, indicating a strong correlation between these variables. The Na, Cl, $HCO_3$, EC,
TDS, and salinity values also had similar patterns to each other. These variables show vertical
symmetry with the planes of Ca, $SO_4$, Sr, and $^{87}Sr/^{86}Sr$. The components of temperature, pH,
DO, K, and $NO_3$ are distinct from each other, showing no relationship with the other
variables.
The clustering could be investigated with the visual inspection of the U-matrix result (Fig.
4). Brown shades on the U-matrix indicate a large distance between neighborhood nodes
whereas white shades correspond to a short distance between nodes. Based on the distances,
the distribution of water samples could be classified into four groups: Group 1 (KW 1, KW 2,
KW 9-1, and KW 10-1), Group 2 (KW 3, KW 5-1, KW 5-2, KW 6-2, KW 11-3, and KW 12-
1), Group 3 (KW 4-1 and KW 4-2), and Group 4 (KW 8-1, KW 11-1, and KW 11-2). This



classification has similar results with the classification based on lithostratigraphic unit data.
Group 1 has relatively high values of Ca, Mg, SO$_4$, NO$_3$, Sr, and $^{87}$Sr/$^{86}$Sr. Group 2 falls
between Group 3 and Group 4. Group 3 is characterized by distinctly high values in K, Na, Cl,
HCO$_3$, EC, TDS, and salinity. Group 4 has high DO values and a distinct low temperature
and $^{87}$Sr/$^{86}$Sr with relatively low values of EC, TDS, salinity, and Sr.
**4.2 Water level, temperature, and EC changes**
Groundwater level changes in seven monitoring wells can be classified into three types: (i)
no change related to the earthquake (KW 5-1 and KW 5-2), (ii) maintenance after an
instantaneous increase or decrease (KW 8-1, KW 8-2, and KW 11-2), and (iii) recovery to
original values after a sudden change (KW 11-1) (Fig. 5). The groundwater level of the KW 5
wells did not change regardless of earthquake and precipitation (Fig. 5a). At the KW 8 wells,
there was an abrupt increase during the earthquake and maintenance after the earthquake,
particularly in the bedrock aquifer well (KW 8-2) (Fig. 5b). The groundwater level response
to the earthquake in the alluvial aquifer well was in contrast to that in the bedrock aquifer
well at the KW 11 wells (Fig. 5c). The groundwater level of KW 11-1 slightly increased
before the earthquake and then decreased. However, the groundwater level of KW 11-2
drastically decreased and then gradually recovered and it remained at higher values compared
to the original values.
Groundwater temperature changes only occurred at the KW 11 wells (Fig. 6a). These also



showed an opposite change pattern, in which the groundwater temperature of KW 11-1
recovered to the original value after an instantaneous increase, whereas that of KW 11-2
recovered after a slight decrease. This anomaly was apparent in the alluvial aquifer well (KW
11-1), unlike the groundwater level anomaly.
A change in groundwater ECs was observed at eight monitoring wells before, during, or
after the earthquake. KW 1 responded to the earthquake in a peak form and gradually
recovered. KW 2 showed an increase prior to the earthquake and recovered to original values.
The groundwater EC consistently decreased and then remained at lower values at KW 6-1,
KW 6-2, and KW 10-2. The peak form of the KW 11 wells also indicated an opposite
direction (Fig. 6b). KW 11-1 peaked at a higher level several times prior to the earthquake,
while KW 11-2 peaked at a lower level before the earthquake and then recovered. Compared
to the water level data, however, it was difficult to interpret that the changes in EC could be
attributed to the earthquake.
**4.3 Radon -222**
The distribution of radon concentration in the 21 groundwater samples is shown in Fig. 7.
The radon concentration ranged from 225 Bq/m$^3$ to 23060 Bq/m$^3$ in the Gyeongju area (see
Table 3). The KW 5-1, KW 5-2, and KW 8-2 values were 15849, 17575, and 23060 Bq/m$^3$,
respectively, which were higher values than those of the other groundwater wells. These wells
are near the epicenter. Lower values (< 1000 Bq/m3) were found in KW6-1, KW 6-2, KW 7-





1, KW 9-1, KW 9-2, KW 10-1, KW 10-2, and KW 11-1. The values between the alluvial and
bedrock aquifer wells were similar in KW 4, KW 5, KW 6, and KW 7. The value difference
between two formation wells was high in KW 8, KW 10, and KW 11. KW 7, KW 9, and KW
11 showed an anomaly in which the alluvial aquifer well had a higher radon concentration
than the bedrock aquifer well.
**4.4 Strontium isotopes**
The strontium isotopic compositions of groundwater samples in the Gyeongju area are
shown in Fig. 8. Strontium concentrations ranged from 18.1 ppb to 4052 ppb. The $^{87}Sr/^{86}Sr$
values ranged from 0.7057 to 0.7124 (see Table 3). In the alluvial aquifer wells, the $^{87}Sr/^{86}Sr$
values ranged from 0.7061 to 0.7083, and these values were from 0.7057 to 0.7124 in the
bedrock aquifer wells. The strontium isotopic compositions of the groundwater samples also
reflected distinct ratios based on their lithology and stratigraphy. The Hayang group (KW 1,
KW 2, KW 9-2 and KW 10-2) had high strontium concentrations. The $^{87}Sr/^{86}Sr$ values of the
Bulguksa group (KW 3, KW 5-2, and KW 12-2) ranged from 0.706 to 0.708. Cretaceous
volcanic rocks (KW 8-2 and KW 11-2) are below the Bulguksa group. The KW 6 wells had
distinct characteristics, in which KW 6-1 was far from KW 6-2.
The spatial distributions of strontium concentrations and $^{87}Sr/^{86}Sr$ are shown in Fig. 9.
Exceptionally high strontium concentrations were observed in KW 1, KW 2, and KW 10-2
(>3000 ppb), whereas KW 3, KW 4-1, KW 6-1, KW 6-2, KW 7-2, KW 8-2, KW 10-3, KW





11-1 had significantly low values (< 100 ppb). The wells that had high strontium
concentrations were in the Hayang group. For the $^{87}Sr/^{86}Sr$ results, KW 1, KW 2, KW 6-2,
KW 10-2, and KW 11-2 had high ratio values, while KW 3, KW 6-1, KW 7-1, KW 8-2, and
KW 11-1 had low ratio values. The values between the alluvial and bedrock aquifer wells
were quite different in KW 6, KW 8, KW 10, and KW 11.
Calcium and strontium cation contents of the groundwater samples showed various
distributions, ranging from 1.59 mg/L to 94.89 mg/L for calcium and from 18.1 ppb to 4052
ppb for strontium concentration (Fig. 10). The positive relationship between Sr and Ca is
consistent with the chemical similarity of strontium and calcium, reflecting similar behavior
in both rock and groundwater. The $Sr^{2+}$ cation contents of the Hayang group ranged from 18.1
ppb to 4052 ppb, which was much higher than the values generally measured in groundwater
as hundreds of ppb (Frost and Toner, 2004;Santoni et al., 2016).

## 5. Discussion

The Gyeongju earthquake on September 12, 2016, remarkably affected the groundwater
systems. The total data showing anomalies are shown in Table 4. The groundwater level,
temperature, and EC data were analyzed considering pre-, co-, and post-seismic changes. For
the groundwater level data, three anomaly types were observed (see Fig. 5). Among them, the
maintenance of a groundwater level increase could be attributed to aquifer compaction (as



observed in KW 8-1 and KW 8-2) (Lee et al., 2002). There is a possibility that the aquifers
underwent non-recoverable deformation. The persistent groundwater level changes also have
been influenced by volumetric strain changes (Matsumoto et al., 2003;Roeloffs et al.,
2003;Wang et al., 2007). In contrast, a greater decrease in groundwater level prior to the
earthquake could be attributed to the opening of bedrock fractures (as observed in KW 11-1)
(Fleeger et al., 1999;Rojstaczer and Wolf, 1992;Rojstaczer et al., 1995). A decrease could also
possibly be related to a change in permeability (Brodsky, 2003;Manga and Wang, 2007).
Groundwater level oscillation also depends on the interaction between the flow in the well
and the flow into and out of the aquifer (Cooper et al., 1965). There is another anomaly, an
opposite change pattern between the alluvial and bedrock aquifer wells, as observed in all
datasets of groundwater level, temperature, and EC data for KW 11. This means that the two
wells had weak interactions with each other.
Isotopic data including radon and strontium were collected only after earthquake. A
difference in radon concentration between the alluvial and bedrock aquifer wells could be
considered more significant because of the mixing effect as observed in KW 8 and KW 11.
Seismotectonic activity may often change the mixing ratio of groundwater in a well
(Hartmann and Levy, 2005). The anomaly in which the alluvial aquifer well had a higher
radon concentration than that of the bedrock aquifer could be attributed to rainfall; however,
in this area, rainfall did not occur during the sampling period (as observed in KW 7, KW 9,
and KW 11). The large variation in the $^{87}Sr/^{86}Sr$ ration in the groundwater can consequently





largely be explained by the nature of the aquifer lithology. For example, the high Rb/Sr ratio
of composite silicate minerals such as plagioclase, feldspar, and biotite can cause granitic
bedrock to be highly radiogenic (Frost and Toner, 2004;Santoni et al., 2016). Generally, the
Bulguksa granite had a strontium concentration from 62 ppm to 428 ppm and an $^{87}$Sr/$^{86}$Sr
ratio from 0.7046 to 0.7114 (Cheong and Jo, 2017). Basaltic rocks near the Yeonil group and
Janggi group had strontium concentration from 439 ppm to 518 ppm and an $^{87}$Sr/$^{86}$Sr ratio
from 0.7039 to 0.7046 (Shimazu et al., 1990). In the Chaeyaksan basaltic volcanics of the
Yucheon group, strontium concentration ranged from 731 ppm to 1667 ppm and the $^{87}$Sr/$^{86}$Sr
ratio from 0.7059 to 0.7064 (Yun, 1998). Thus, the more radiogenic samples of the Bulguksa
granite were expected compared to those of the Yucheon group rocks, because of the high
biotite content of the Bulguksa granite.
The most significant and novel result from this study is that these responses were analyzed
with the result of grouping conducted using the SOM statistic tool. The SOM results were in
agreement with the lithostratigraphic unit data which was useful in arranging the bedrock
aquifer wells, based on bedrock characteristics. The final grouping yielded four classes of
wells: Group A (KW 1, KW 2, KW 9, and KW 10); Group B (KW 3, KW 5, and KW 12);
Group C (KW 4, KW 6, and KW 7); and Group D (KW 8 and KW 11). This grouping was
conducted as one well binding the alluvial and bedrock aquifer wells.
The lithology and stratigraphy of Group A is classified as Hayang group shale and sandstone





of low porosity and high strontium concentrations. Particularly, the KW 9 and the KW 10
wells had a low radon concentration (< 1000 Bq/m3), high strontium concentration, and high
Ca value (see Fig. 10). There might be some possible mechanisms for the exceptionally
strong chemical signatures. Regarding earthquakes, first, the fine-grained bedrock of Group A
has a large reactive surface area that can effectively activate water–rock interaction and
largely vary the groundwater chemistry via ion exchange (Pennisi et al., 2006). These
interactions can cause high strontium concentrations. Second, particularly for KW10-2 in
Group A, the exceptionally high Sr samples appear to be an effect of cation exchange
between the soil and surrounding water. The capacity of the cation-bearing soil (cation-
exchange capacity; CEC) depends on the pH of the surrounding water, and $Ca^{2+}$ and $Sr^{2+}$ are
characterized by particularly high replaceability (Carroll, 1959). The acidic water of KW 10-
2 (pH = 2.27) would lead to a lower CEC in the soil and dissolution of $Ca^{2+}$ and $Sr^{2+}$ from the
soil grain surface as hydrogen ions replace Ca and Sr in the soil. The flow into the
groundwater in the Hayang group rocks could increase the chemical concentration of Group
A. Third, the results could be attributed to geological characteristics, not related to the
earthquake, as the intrinsic chemistry of the Hayang group shale and sandstone might affect
the strontium concentrations. Such dramatically high values of Sr were previously observed
in the Redbeds aquifer (885–7851 ppb) where the lithology of the bedrock is composed of
shale and sandstone with high Rb/Sr ratios (Santoni et al., 2016).
Group B wells are located in a granitic biotite region of the Bulguksa group, which has a



typical high radon concentration. The radon concentration is greatly influenced by uranium
content; thus, its concentration is generally high in granite compared to that of sedimentary
rocks. Typically, uranium concentration is high in granites, whereas it is low in sedimentary
rocks. However, only the KW 5 wells had a high radon concentration. In particular, KW 5-1
had high values similar to those of KW 5-2. This could be attributed to deep fluid upwelling
from the bedrock in the KW 5 wells (Chiodini et al., 2000;Minissale, 2004;Savoy et al., 2011).
However, it is difficult to confidently determine an effect of upwelling because data were
only collected after the earthquake, not prior.
Group C is composed of tuff and tuffaceous sedimentary rocks of the Yeonil and Janggi
groups. This group had a low radon concentration and a small difference in radon
concentration between the alluvial and the bedrock aquifer wells (see Fig. 7), suggesting
active water mixing between the two aquifers. In addition, the bedrock of this area contains
conglomerates, which generally have high pore density, leading to active mixing with water
compared to the shale-dominant lithology. This hypothesis seems to be consistent with the
weak chemical signature of Group A. Moreover, KW 4-1, KW 4-2, and KW 6-2 had high
values of EC and Cl, suggesting the possibility of sea water intrusion in the wells. Sea water
intrusion might actively trigger mixing between the shallow and deep aquifers. The strontium
concentration and Ca values are also low in these wells (see Fig. 10).
In Group D, the radon concentration was quite different between the two wells and a





groundwater level anomaly occurred (see Fig. 7). The wells of this group are in Cretaceous
mainly andesitic volcanic rocks. The KW 11 wells, in particular, showed many factors
including groundwater level, temperature, and EC responded to the earthquake in an opposite
manner (see Fig. 5 and Fig. 6). The radon concentration of KW 11-1 was also higher than that
of KW 11-2. The Sr contents of Group B and Group D show a wide range of concentrations
observed in other studies of groundwater in the granitic bedrock aquifers; e.g., an $Sr^{2+}$ from
67 to 169 ppb (Frost and Toner, 2004) and from 103 to 553 ppb (Santoni et al., 2016). This
wide range might be associated with the different amount of plagioclase feldspar in each
matrix rock of the groundwater. Water flow via granite can be controlled by the dissolution of
anorthite and alkali feldspar. The former occurs more rapidly, providing $Ca^{2+}$ and $Sr^{2+}$ with a
low $^{87}Sr/^{86}Sr$ ratio (Bullen et al., 1997;Franklyn et al., 1991). In contrast, one groundwater
chemistry study in Canada showed that dissolution of alkali feldspar can increase the
$^{87}Sr/^{86}Sr$ ratio providing sodium and potassium (Bullen et al., 1996). Therefore, the various
compositions of the granite and the fluid mobility would be determinative in the $^{87}Sr/^{86}Sr$
ratio.
In accordance with this analysis, conceptual models of groundwater changes induced by the
earthquakes can be suggested (Fig. 11). Four different models were inferred by data analysis
and the grouping result using the SOM approach. First, a response highlighting the mixing
with deep groundwater or bedrock can be attributed to a deep fluid rise, which resulted in
high strontium concentrations, as observed in the wells of Group A (KW 1 and KW 2). In



addition, low radon values and high $^{87}$Sr/$^{86}$Sr ratios were observed in the wells of the alluvial
aquifer, KW 9-1 and KW 10-1. Second, the possibility of non-recoverable deformation after
deep fluid upwelling can be suggested as there was no change in water level and there were
high radon concentrations in both wells of the alluvial and bedrock aquifers (KW 5-1 and
KW 5-2) in Group B. This hypothesis can be supported by studies showing that the stress
reduction after an earthquake causes closure of cracks (Nur and Booker, 1972;Scholz et al.,
1973). The other wells of Group B could be classified as an uninfluenced by the earthquakes.
Third, another mechanism, the strong interaction between shallow and deep aquifers, can be
attributed to sea water intrusion by the data showing a small difference in radon concentration
between the alluvial and the bedrock aquifer wells, as observed in Group C. Finally, the
response to the movement of the strike-slip fault can be explained considering the location of
Group D, which is near the YSF. The water level anomaly suggests the potential that the
source of the alluvial aquifer well changed a different source compared to that of bedrock
aquifer well after the earthquakes (see Fig 5). Bedrock fracture opening could cause a
decrease in water level, suggesting that surrounding aquifer affected the alluvial aquifer of
these wells because of the difference in water level, as observed in KW 11-2. In contrast, the
groundwater level appeared to remain constant at a higher value than the pre-earthquake
value via aquifer compaction because of the movement of the strike-slip fault at the KW 8
wells.
This conceptual model should, however, be augmented and validated by more detailed




hydrogeological characterizations because of the limited dataset. Further monitoring or
modeling works will help to reinforce the proposed model.

**6. Conclusion**
The 2016 Gyeongju earthquakes affected the pre-, co-, and post-earthquake groundwater
systems. Changes were observed in groundwater level, temperature, EC, hydrochemistry,
radon-222, and strontium isotopic data. The main findings obtained via data analysis from 21
monitoring wells are as follows:
1.   The observed groundwater level anomaly could be attributed to pre-earthquakes effect,

not a seasonal effect. Maintenance, persistent or abrupt changes, and oscillation of

water levels were observed in some wells.

2.   The radon concentration could be interpreted as the difference between alluvial and

bedrock aquifer wells. A relatively small difference between two radon values implies

active mixing processes between the shallow and deep aquifers.

3.   Strontium isotopes were interpreted with the lithology and stratigraphy of bedrock,

indicating the potential of water–rock interactions. These isotopes ($Sr^{2+}$ concentrations

and $^{87}Sr/^{86}Sr$ ratio) also could suggest both geologically independent causes and

dependent causes with respect to the earthquakes.



4.  The SOM statistic tool was found to be useful for identifying each group having

common characteristics and the influence of the earthquakes on hydrogeochemical

parameters. The final grouping can explain the possible mechanisms via different

hydrogeochemical processes: (i) water–rock interactions because of deep fluid rising, (ii)

no response to the earthquakes or non-recoverable deformation after the earthquake, (iii)

aquifer mixing vertically due to sea water intrusion, and (iv) the effect of the movement

of the strike-slip fault.

These results can have significant impact on regional and national Authorities, because
seismicity has increased in the area near Gyeongju since 2016. It may be more helpful in
efficiently managing groundwater systems to analyze the combined hydrogeochemical and
lithostratigraphic characteristics of the area. In addition, the studied parameters and the
adopted methods would be positively applied for other earthquake zones, particularly for
grouping interpretation of response of monitoring wells.

**7.  Data availability**
The dataset for water level, temperature, and electrical conductivity presented in this paper is
available online at http://www.gims.go.kr. The precipitation data is available at
http://kma.go.kr.





**8.  Author contribution**
J. Kim and K.K. Lee had the idea and supervised this paper. J. Lee wrote the geological
setting of the study area and drew some figures. M.P. discussed the results and contributed to
writing the paper. All authors designed sampling method and analyzed the samples.

**9.  Competing interests**
The authors declare that they have no conflict of interest.











**Acknowledgments.**
This work was supported by the National Research Foundation of Korea (NRF) grant funded
by the Korea government(MSIP) (No. 2017R1A2B3002119)

















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

**marked by circles illustrated by the color table. Gyeongju (yellow square) and the well**
**locations (blue squares) are highlighted. (b) Geological map of the study area. The color**
**legend shows the lithostratigraphic units comprising the Gyeongju area. Major faults**
**comprising the Yangsan Fault System are denoted with abbreviations; YSF, Yangsan**
**Fault; MoRF, Moryang Fault, MiRF, Miryang Fault; USF, Ulsan Fault, JNF, Jain Fault.**
**Figure. 3. Visualization of the component planes of the hydrogeochemical data for the**
**Gyeongju area from the SOM results.**
**Figure. 4. U-matrix visualization and pattern of group formation of the SOM results in**
**the Gyeongju area.**
**Figure. 5. Time series data of groundwater level in (a) KW 5; (b) KW 8; and (c) KW 11.**
**The dates of the mainshock and aftershock of the earthquake ($M_L \geq 4.5$) are marked as**
**the orange colored line.**





**Figure. 6. Time series data of the KW 11 well: (a) temperature and (b) electrical conductivities. The dates of the mainshock and aftershock of the earthquake ($M_L \geq 4.5$) are marked as the orange colored line.**

**Figure. 7. Spatial distribution of radon concentrations in the Gyeongju area.**

**Figure. 8. $^{87}Sr/^{86}Sr$ vs 1/Sr plot for the groundwater samples. The rectangular boxes indicate each group defined considering the results of both SOM and lithostratigraphy. Green colored box is Group A (shale and sandstone), orange colored box is Group B (granite), yellow colored box is the KW 6 wells of Group C, and the red colored box is Group D (andesite).**

**Figure. 9. Spatial distribution of strontium concentrations and $^{87}Sr/^{86}Sr$ ratios in the Gyeongju area.**

**Figure. 10. Correlation plot of strontium and calcium values of the groundwater samples in Gyeongju area.**

**Figure. 11. (a) Conceptual model to explain the responses of the groundwater system induced by the Gyeongju earthquakes: active water-rock interactions increasing the geochemical signature (KW 9 in Group A), water level anomaly related to non-recoverable deformation (KW 5 in Group B) (dotted line indicates the water table before the earthquakes, the solid line and red inverted triangle indicate the water table after the earthquakes), strong mixing between shallow and deep aquifer caused by sea**



**water intrusion (KW 4 in the Group C), and strike-slip deformation leading to the**
**difference between the alluvial aquifer and the bedrock aquifer (Group D). (b)**
**Simplified geological cross section of KW 6-1.**















**Table 1. The mainshock and aftershocks data (ML≥3.5) of the Gyeongju earthquake.**

| Date, time | $M_L$ | Longitude | Latitude |
|---|---|---|---|
| 2016-11-13, 21:52:57 | 3.5 | 36.36 N | 126.63 E |
| 2016-11-06, 06:26:22 | 3.5 | 33.76 N | 125.07 E |
| 2016-09-21, 11:53:54 | 3.5 | 35.75 N | 129.18 E |
| 2016-09-19, 20:33:58 | 4.5 | 35.74 N | 129.18 E |
| 2016-09-12, 20:34:22 | 3.6 | 35.78 N | 129.19 E |
| *2016-09-12, 20:32:54* | *5.8* | *35.76 N* | *129.19 E* |
| *2016-09-12, 19:44:32* | *5.1* | *35.77 N* | *129.19 E* |

[†] The bold italics is the mainshock of the Gyeongju earthquakes.



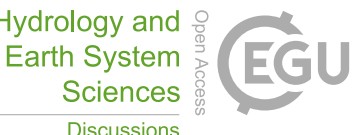

**Table 2. Groundwater well information.**

| Well ID | Longitude | Latitude | Distance from epicenter (km) | Well type | Lithostratigraphic unit | Sampling depth (m) |
|---|---|---|---|---|---|---|
| KW 1 | 36.17 N | 128.72 E | 59.99 | Bedrock | Hayang Group (cretaceous grey, dark grey siltstone and shale) | 30 |
| KW 2 | 36.11 N | 128.92 E | 45.85 | Bedrock | Hayang Group (cretaceous greenish grey, dark grey shale with carbonate and sandstone) | 50 |
| KW 3 | 36.13 N | 129.26 E | 41.82 | Bedrock | Bulguksa Granite (biotite granites of Late Cretaceous to Early Tertiary age) | 20 |
| KW 4 | 36.00 N | 129.31 E | 29.33 | Alluvial, Bedrock | Yeonil Group (light brown shale and mudstone coexisting with conglomerate of Miocene age) | 28, 35 |
| KW 5 | 35.75 N | 129.32 E | 12.57 | Alluvial, Bedrock | Bulguksa Granite (biotite granites of Late Cretaceous to Early Tertiary age) | 8, 39 |
| KW 6 | 35.83 N | 129.41 E | 21.51 | Alluvial, Bedrock | Janggi Group (andesite and tuff of Miocene age) | 20, 30 |
| KW 7 | 35.90 N | 129.27 E | 16.94 | Alluvial, Bedrock | Yeonil Group (conglomerate of Miocene age) | 5, 50 |
| KW 8 | 35.75 N | 129.05 E | 12.76 | Alluvial, Bedrock | Yucheon Group (andesite, porphyry andesite, and brecciated andesite of Cretaceous age) | 5, 40 |
| KW 9 | 35.82 N | 129.10 E | 10.57 | Alluvial, Bedrock | Hayang Group (black and greenish grey shale with hornfels of Cretaceous age) | 10, 50 |
| KW 10 | 35.58 N | 129.21 E | 20.13 | Alluvial, Bedrock | Hayang Group (greenish grey and dark grey sandstone, siltstone, shale coexisting with mudstone and conglomerate of Cretaceous age) | 10, 28 |
| KW 11 | 35.62 N | 129.08 E | 19.35 | Alluvial, Bedrock | Yucheon Group (granite of Cretaceous age) | 10,28 |
| KW 12 | 35.75 N | 128.65 E | 48.68 | Alluvial, Bedrock | Bulguksa Granite (intrusive rocks and granite porphyry of Cretaceous age) | 8, 50 |

[†]Well type refers the aquifer characteristics of location of installed well. The alluvial refers the well installed in the alluvial aquifer.



**Table 3. Hydrogeochemical data collected after 2016 Gyeongju earthquake.**

| Well ID | Ca | K | Mg | Na | Cl | SO$_4$ | NO$_3$ | HCO$_3$ | Sr | Tem. | pH | DO | EC | TDS | Sal. | Radon | $^{87}$Sr/$^{86}$Sr |
|---|---|---|---|---|---|---|---|---|---|---|---|---|---|---|---|---|---|
| | | | | (mg/L) | | | | | (ppb) | (℃) | | (mg/L) | (μs/cm) | (mg/L) | (%) | (Bq/m$^3$) | |
| KW 1 | 86.0 | 2.27 | 33.7 | 63.1 | 13.6 | 247 | 0.52 | 292 | 3660 | 14.8 | 7.03 | 0.72 | 898 | 0.58 | 0.44 | 6693 | 0.712368 |
| KW 2 | 48.7 | 1.31 | 15.9 | 21.0 | 15.5 | 34.6 | 15.9 | 182 | 3393 | 15 | 7.59 | 0.95 | 412.1 | 0.27 | 0.2 | 3193 | 0.709754 |
| KW 3 | 13.8 | 0.56 | 3.27 | 51.8 | 32.5 | 7.92 | 13.1 | 91.2 | 31.3 | 15.5 | 7.87 | 3.16 | 294.2 | 0.19 | 0.14 | 2366 | 0.706575 |
| KW 4-1 | 8.91 | 16.9 | 11.8 | 193 | 111 | 73.8 | 0.21 | 300 | 79.6 | 15 | 8.45 | 0.62 | 950 | 0.62 | 0.47 | 2416 | 0.708188 |
| KW 4-2 | 6.32 | 6.33 | 7.78 | 778 | 721 | 45.7 | 0.25 | 846 | 225 | 15.9 | 8.26 | 0.66 | 3310 | 2.15 | 1.74 | 2425 | 0.707283 |
| KW 5-1 | 17.7 | 2.67 | 7.07 | 13.0 | 9.45 | 49.5 | 13.5 | 31.2 | 147 | 15.4 | 7.52 | 2.24 | 213.1 | 0.14 | 0.1 | 15849 | 0.707610 |
| KW 5-2 | 30.5 | 2.43 | 5.11 | 35.2 | 17.7 | 27.9 | 0.34 | 137 | 170 | 15.6 | 6.55 | 0.74 | 309.7 | 0.2 | 0.15 | 17575 | 0.707356 |
| KW 6-1 | 1.76 | 3.79 | 0.99 | 74.4 | 21.6 | 26.0 | 0.28 | 124 | 18.1 | 15 | 6.87 | 0.64 | 315.5 | 0.21 | 0.15 | 225 | 0.706191 |
| KW 6-2 | 1.59 | 1.75 | 0.68 | 146 | 33.9 | 17.0 | 0.14 | 257 | 65.1 | 14.9 | 8.15 | 0.69 | 530 | 0.34 | 0.26 | 368 | 0.711835 |
| KW 7-1 | 15.8 | 0.91 | 5.75 | 15.7 | 11.1 | 2.47 | 1.15 | 90.2 | 117 | 16.1 | 7.29 | 0.8 | 180.1 | 0.12 | 0.09 | 1218 | 0.706590 |
| KW 7-2 | 9.94 | 0.99 | 3.71 | 15.8 | 9.62 | 2.79 | 1.95 | 68.3 | 78.0 | 15.1 | 7.01 | 2.76 | 140.3 | 0.09 | 0.07 | 992 | 0.705688 |
| KW 8-1 | 19.1 | 6.36 | 3.91 | 19.2 | 22.2 | 16.8 | 15.9 | 57.7 | 115 | 11.5 | 7.01 | 1.63 | 224.7 | 0.15 | 0.11 | 5974 | 0.708231 |
| KW 8-2 | 12.7 | 0.33 | 2.02 | 18.5 | 3.52 | 15.3 | 0.81 | 64.4 | 75.0 | 14.6 | 7.3 | 0.82 | 146.8 | 0.03 | 0.02 | 23060 | 0.706177 |
| KW 9-1 | 53.6 | 16.9 | 21.6 | 12.3 | 18.4 | 46.6 | 40.9 | 187 | 379 | 14.3 | 7.02 | 5.17 | 513 | 0.33 | 0.25 | 585 | 0.707919 |
| KW 9-2 | 62.4 | 10.4 | 24.0 | 13.3 | 18.6 | 43.8 | 38.9 | 204 | 538 | 14.9 | 6.9 | 0.58 | 521 | 0.34 | 0.25 | 249 | 0.707469 |
| KW 10-1 | 29.2 | 4.23 | 6.50 | 21.0 | 22.5 | 31.3 | 19.2 | 76.6 | 194 | 17.2 | 7.69 | 7.05 | 294.7 | 0.19 | 0.14 | 228 | 0.708353 |
| KW 10-2 | 94.9 | 2.27 | 13.2 | 92.6 | 15.5 | 305 | 10.3 | 161 | 4052 | 16.3 | 2.27 | 7.07 | 865 | 0.56 | 0.43 | 758 | 0.712029 |
| KW 11-1 | 11.7 | 2.11 | 1.93 | 8.38 | 7.04 | 13.1 | 12.8 | 32.6 | 82.6 | 9.4 | 7.53 | 11.26 | 215.9 | 0.14 | 0.1 | 4204 | 0.706385 |
| KW 11-2 | 32.6 | 1.49 | 2.19 | 13.0 | 7.65 | 34.6 | 4.14 | 79.6 | 54.2 | 14.9 | 7.21 | 6.98 | 121.5 | 0.08 | 0.06 | 488 | 0.706122 |
| KW 11-3 | 24.4 | 2.23 | 8.02 | 7.88 | 7.87 | 14.6 | 7.87 | 93.6 | 212 | 15 | 7.07 | 0.96 | 227.3 | 0.15 | 0.11 | 1950 | 0.709625 |
| KW 12-1 | 20.9 | 5.02 | 5.61 | 15.4 | 12.9 | 34.7 | 18.7 | 51.0 | 134 | 15 | 6.34 | 3.81 | 231.6 | 0.15 | 0.11 | 1755 | 0.707885 |
| KW 12-2 | 24.0 | 6.19 | 5.22 | 15.1 | 13.6 | 32.2 | 22.9 | 58.9 | 136 | 14.8 | 6.11 | 4.52 | 247.6 | 0.16 | 0.12 | 1088 | 0.708022 |

[†]KW ##-1 refers the alluvial aquifer well and KW ##-2 or no hyphen well refers the bedrock aquifer well.



**Table 4. Anomaly data of groundwater wells based on the grouping results.**

| Group | Well ID | | Well type | Groundwater level | Temperature | EC | Radon con. (H : > 15000 L : < 800) | Radon con. Difference | Strontium con. (H : > 3000 L : < 100) | $^{87}$Sr/$^{86}$Sr (H : > 0.709 L : < 0.707) |
|---|---|---|---|---|---|---|---|---|---|---|
| A | KW 1 | | Bedrock | | | O | | | H | H |
| | KW2 | | Bedrock | | | O | | | H | H |
| | KW 9 | KW 9-1 | Alluvial | | | | L | | | |
| | | KW 9-2 | Bedrock | | | | L | | | |
| | KW 10 | KW 10-1 | Alluvial | | | | L | | | |
| | | KW 10-2 | Bedrock | | | O | L | | H | H |
| B | KW 3 | | Bedrock | | | | | | L | L |
| | KW 5 | KW 5-1 | Alluvial | O | | | H | | | |
| | | KW 5-2 | Bedrock | O | | | H | | | |
| | KW 12 | KW 12-1 | Alluvial | | | | | | | |
| | | KW 12-2 | Bedrock | | | | | | | |
| C | KW 4 | KW 4-1 | Alluvial | | | | | L | L | |
| | | KW 4-2 | Bedrock | | | | | L | | |
| | KW 6 | KW 6-1 | Alluvial | | | O | L | L | L | L |
| | | KW 6-2 | Bedrock | | | O | L | L | L | H |
| | KW 7 | KW 7-1 | Alluvial | | | | | L | | L |
| | | KW 7-2 | Bedrock | | | | L | L | L | |
| D | KW 8 | KW 8-1 | Alluvial | O | | | | H | | |
| | | KW 8-2 | Bedrock | O | | | H | H | L | L |
| | KW 11 | KW 11-1 | Alluvial | O | O | O | L | H | L | L |
| | | KW 11-2 | Bedrock | O | O | O | | H | | H |

† 'O' refers that the anomaly was detected, 'H' refers the high concentration, and 'L' refers the low concentration.





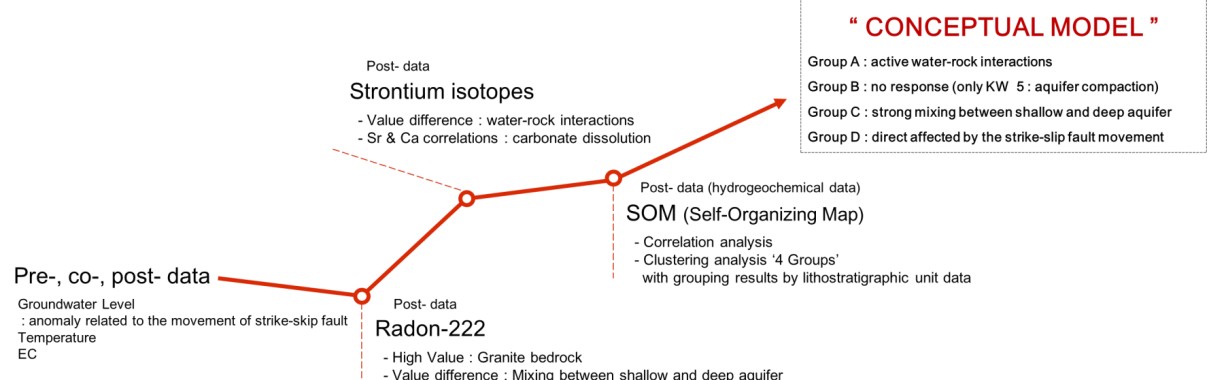

**Figure. 1**



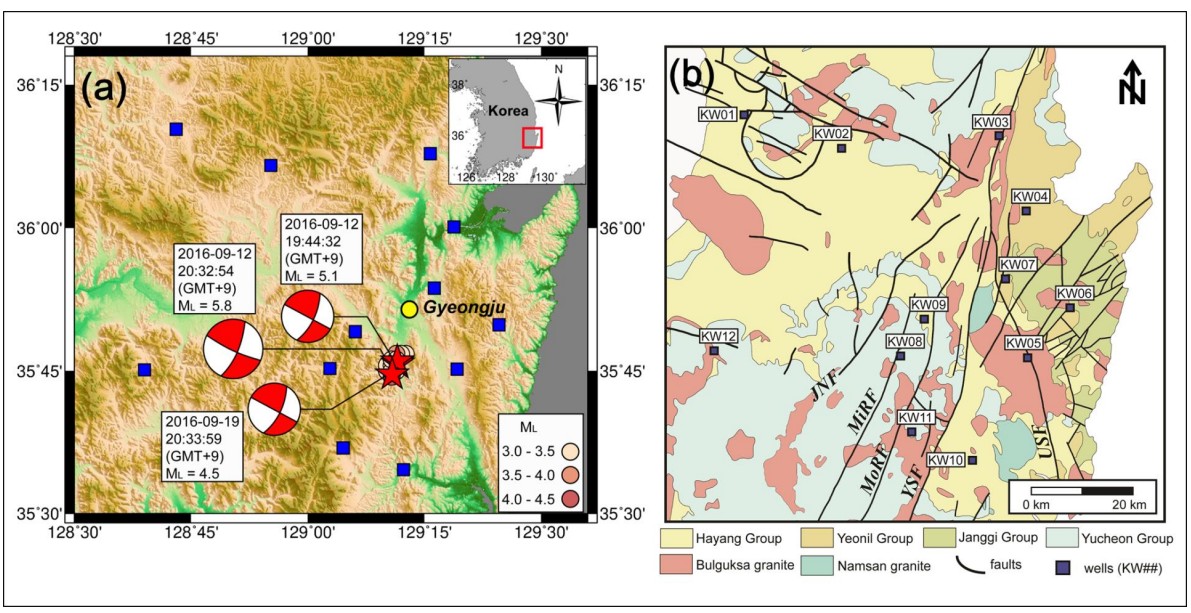

Figure. 2

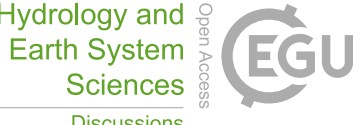

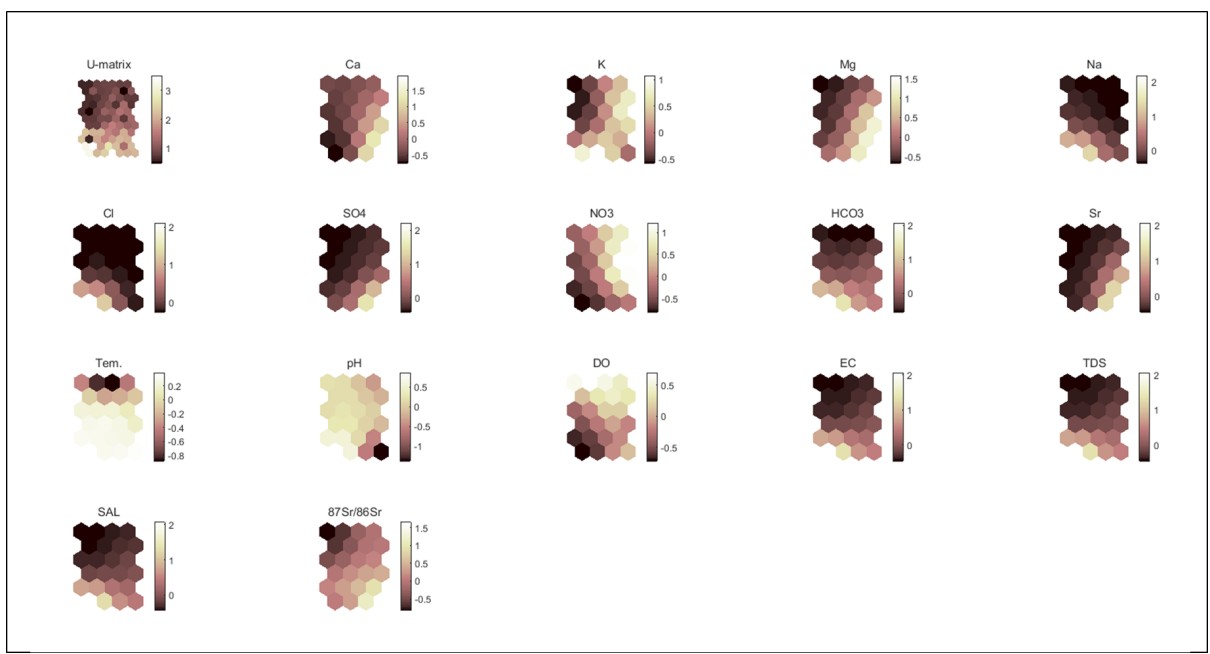

Figure. 3

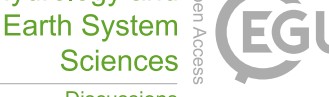



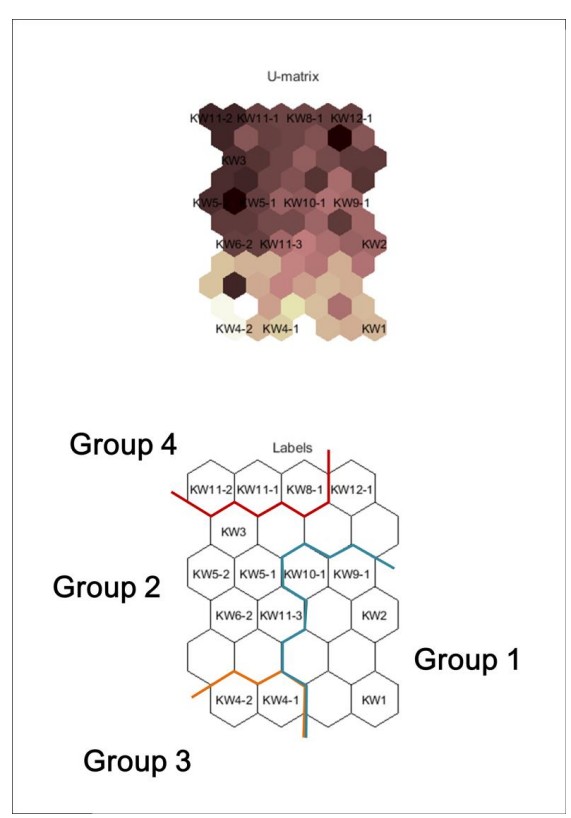

**Figure. 4**





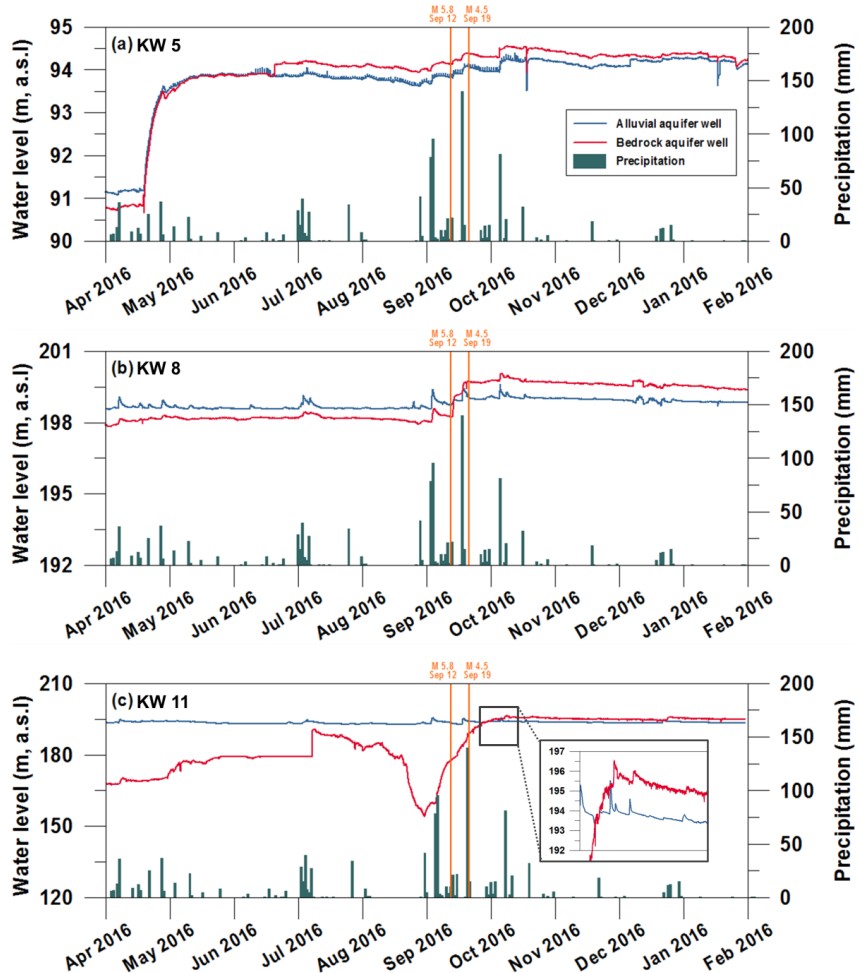

**Figure. 5**



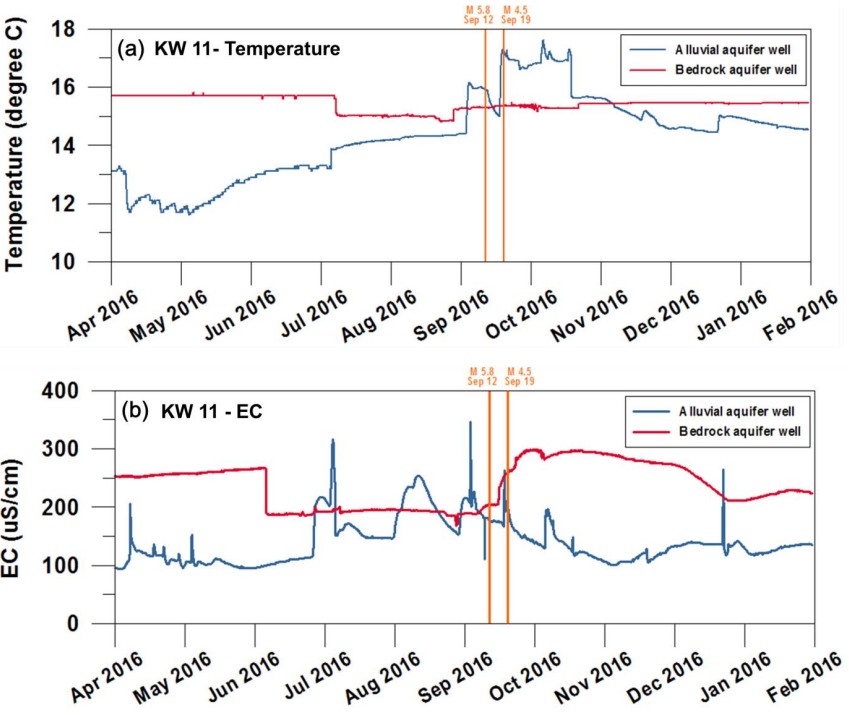

**Figure. 6**





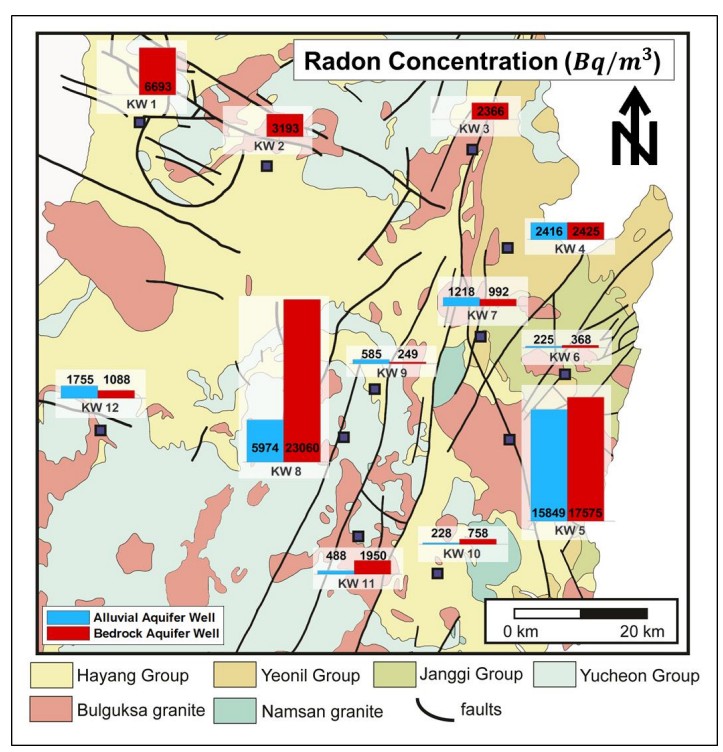

**Figure. 7**



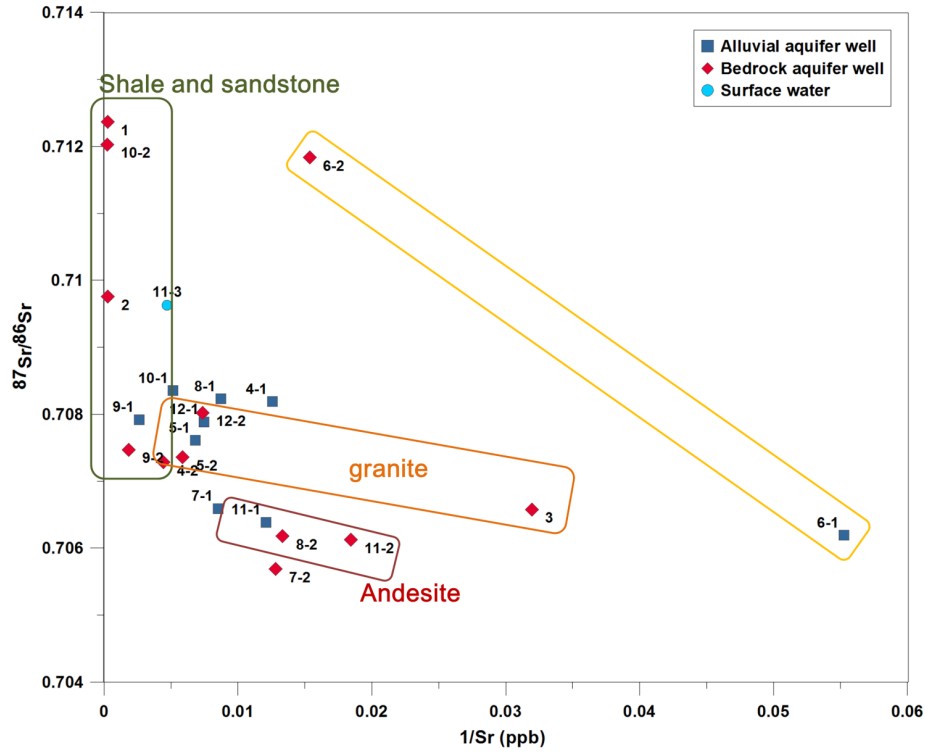

**Figure. 8**





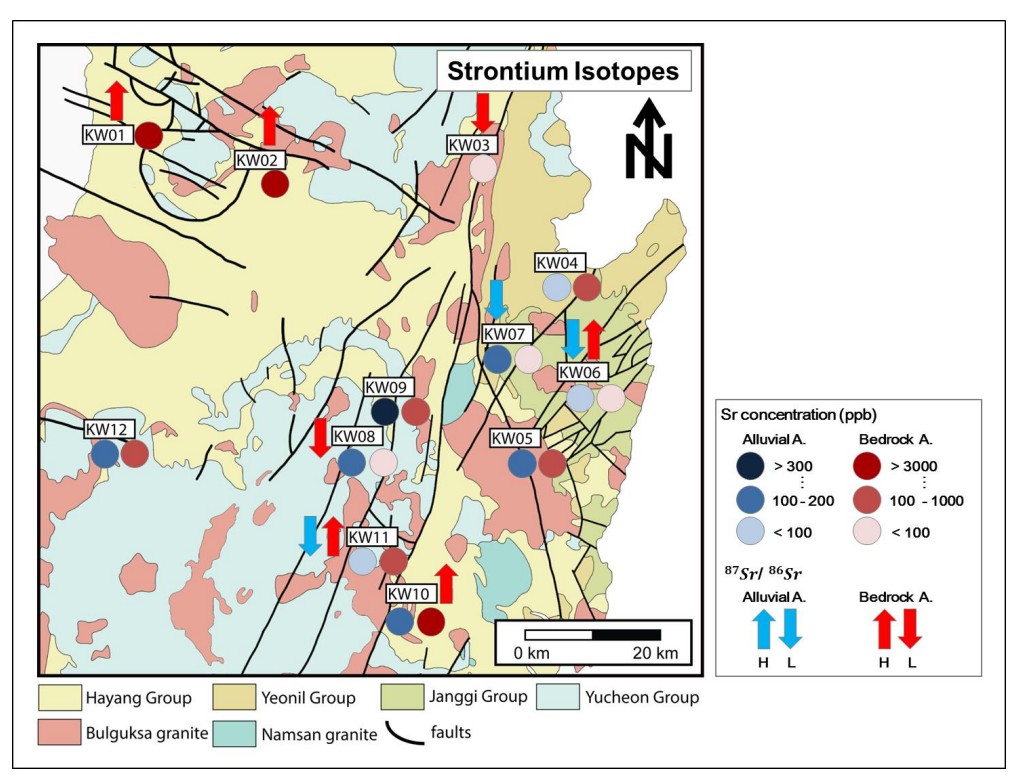

**Figure. 9**



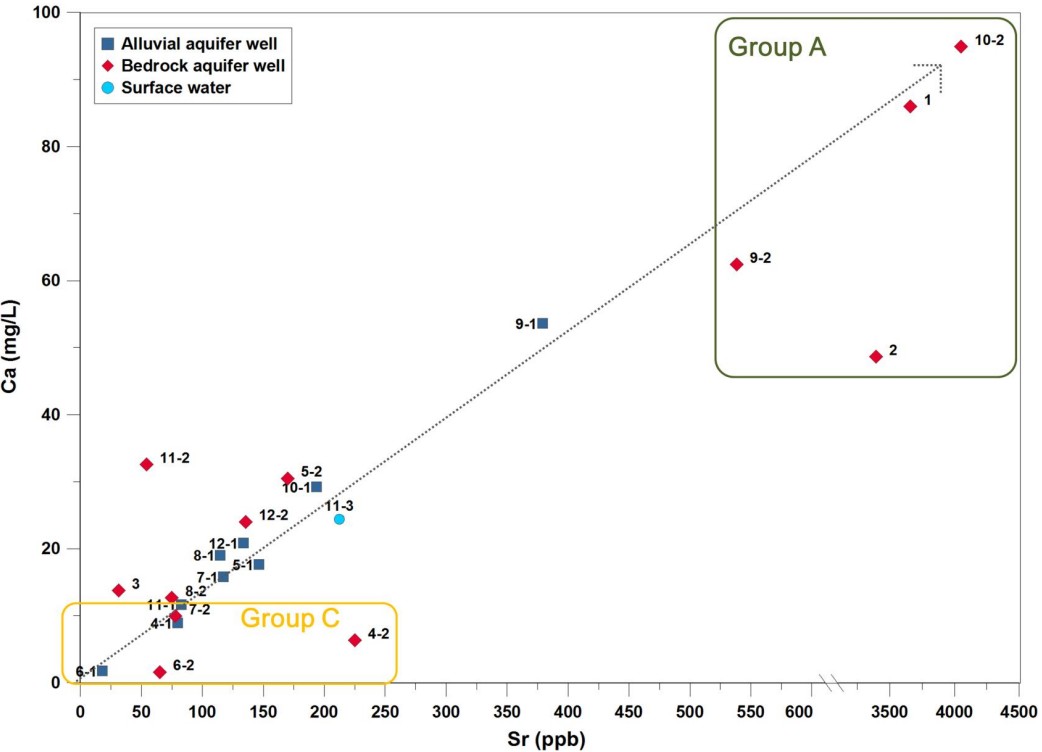

**Figure. 10**





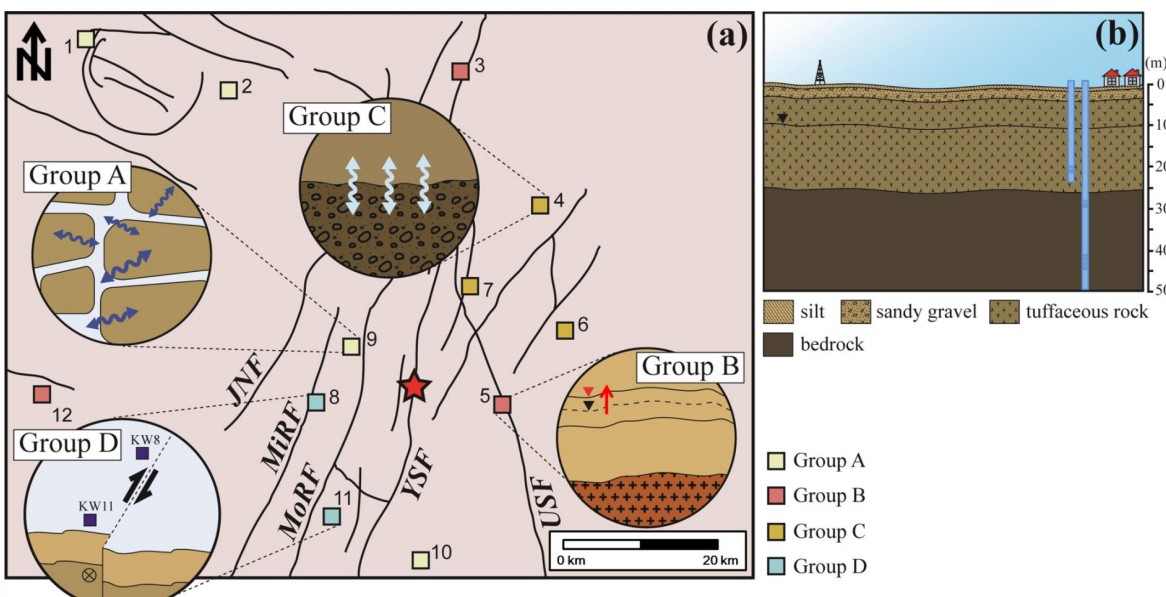

**Figure. 11**