# Peer review of "Hydrogeological responses to the 2016 Gyeongju earthquakes,"

_Hydrology and Earth System Sciences, 2018_

## Referee Comment (RC1) · Anonymous Referee #1 · 27 Aug 2018

**1   General comments**

The manuscript is a technical report which provides an interesting analysis of hydrological data and discusses the evolution of hydrological quantities before, during and after some earthquakes in South Korea.

The work is largely based on well-established methods and the scientific novelty is not very remarkable. The main goal is stated in the introduction at lines 105 to 107 and then later at lines 357 & 358. However, the neural network method applied in this paper (SOM - Self-Organizing Map) is not new and it is not apparent its advantage with respect to other clustering methods. See also the specific comment # 4.

The manuscript is generally well organized and written, but it requires linguistic improvements, some of which are listed in the technical comments below.

From the scientific point of view, at some points, the manuscript is not sufficiently precise and rigorous, as described, e.g., in the specific comments # 1, 2 & 3 and in the technical comments # 9 & 10 below.

I am sorry, but I think that the innovative character of the work is not sufficient to publish the paper on HESS, whereas the manuscript is more adequate for a strictly hydrogeological journal.

**2 Specific comments**

1. Lines 37 to 41. The list of references shows that it is debatable to state that "few" studies are devoted to this research topic. A simple and fast search on google scholar shows a lot of papers related to the effects of earthquakes on hydrological processes and quantities. Perhaps, the Authors want to stress that most papers are devoted to earthquake precursors or to the study of co-seismic phenomena.

2. Throughout the paper, it would be necessary to consider in a more accurate and rigorous way the considered time scales. The following remarks provide some instances.

    - Lines 43 & 44. If the changes are related to seismic waves, they should disappear after the earthquake. Effects at different time-scales should be separated more clearly. The sentence "Seismic waves... geochemistry" should be better connected with the preceding one "Seismicity... groundwater systems".
    - Line 187. The time scales should be considered in a more accurate and rigorous way. In fact, the sampling period of hydrological data is high with

respect to the duration of the earthquake wave train.

3. Line 93. Surface area is an extensive property: does radon concentration (which is an intensive property) depend on it? Should "surface area" be substituted with the intensive property "specific surface"?

4. In section 5 it is shown that several different processes might explain the behaviour of collected data. However, most (if not all) of such processes have been hypothesized in previously published papers: many of the papers cited in this section were published in the second half of the XX century. Such a discussion has a great value for local land management and natural risks mitigation, but a more limited interest for the international scientific community. Moreover, the declared goal of the paper is to show the relevance of the use of SOM, but the discussion of the relevance of this method – as compared with other possible approaches – is almost absent.

5. Carefully revise the number of significant digits used for several quantities. For instance, with reference to Table 3, is it physically significant to express concentrations with three or four significant digits, radon activity with up to five significant digits and Sr isotopic ratio with six significant digits?

**3  Technical comments**

1. Line 9. Specify $M_L$.

2. Lines 15 & 16. Rephrase "with bedrock characteristics".

3. Lines 16 to 19. Erase "To analyze... from the earthquakes" and rephrase the remaining part of the sentence "annual monitoring data... during January 2017".

4. Line 35. Substitute "underground" with "groundwater".

5. Line 55. Rephrase "By using hydraulic properties".

6. Line 61. Rephrase "was recorded as the largest".

7. Line 64 to 69. The sentences "The occurrence... near the YSF (Lee and Jin, 1991;Lee and Na, 1983)" could be erased, since they do not give any scientific information relevant for the paper's objectives.

8. Line 69. Substitute "interpreted" with "shown".

9. Lines 72 to 75. The sentence "The occurrence... following the Gyeongju earthquakes" is quite self-evident and could be better rephrased.

10. Lines 81 & 82. Please explain how temperatures can be derived from measurements of groundwater level.

11. Lines 97 & 98. Add references.

12. Line 99. Specify "these": strontium only or radon and strontium?

13. Line 101. Rephrase "according to the rock type in the bedrock of aquifers"

14. Line 129. Substitute "is approximately 21–35 km" with "varies between 21 km and 35 km".

15. Lines 235 to 237. Rephrase, possibly as "The raw data were normalized in order to work with transformed quantities with zero mean and unit standard deviation".

16. Line 238. Substitute "have" with "show".

17. Line 287. Correct the exponent in $Bq \cdot m^{-3}$.

18. Line 333 & 334. Rephrase the sentence "Groundwater level oscillation... of the aquifer".

19. Line 355. Rephrase "were expected compared to those".

20. Line 359. Rephrase "which was useful".

21. Line 363. Rephrase "as one well binding the alluvial and bedrock aquifer wells".

22. Line 382. Substitute "885–7851 ppb" with "from 885 ppb to 7851 ppb".

23. Line 476. What about hydrogeochemical data?

24. Line 765. Use capital letters for "C.-Y.".

25. Table 3. Rephrase the text of the footnote. Moreover, information is given for KW *-1 and KW *-2: what about KW 11-3? I am afraid that further details are missing: are the screened intervals located at different depths, or are these clusters of wells with different depths?

26. Figure 1. Sorry, but I do not understand this Figure. A more accurate and thorough description in the figure caption is necessary.

27. Figure 2. The colour scale of map (a) is a representation of ground surface level, isn't it? Maps (a) and (b) have the same extension, haven't they? Add this information in the figure caption and the colour scale bar in map (a).

28. Figures 3 & 4. These representations are not easily interpreted. More details in the text and in the figure captions could be useful.

---

## Referee Comment (RC2) · Anonymous Referee #2 · 11 Oct 2018

Comments on the Kim et al. Paper

The manuscript submitted for publication by Kim et al., in HESS seeks to evaluate the impact of seismic events on groundwater dynamics and geochemistry, in the case of the September 12, 2016 Gyeongju earthquakes in Korea.

This scientific issue is of importance and in the scope of the scientific themes published in HESS. The work is based on an annual monitoring of groundwater level, temperature and electro-conductivity of several wells in the geographical area impacted by the seism. Such a monitoring should allow the authors to study the variations of the above parameters before, during and after the earthquakes, which should be at the heart of a discussion on the potential hydrological modifications linked to the seismic events, which is actually not the case. The authors rather based their discussion on geochemical data, including Sr isotope ratios and Rn data, of water samples collected in January 2017, after the seism and on a statistical analysis ("Self-Organizing Map (SOM) " ) of the hydro-geochemical characteristics of the groundwaters.

Except if I have misunderstood the manuscript, I do not really understand such a choice, and I have many difficulties to really understand the arguments developed by the authors to sustain/defend the interpretations given in the discussion. The discussion under its present form is based on many general considerations on the origin of Sr isotope ratios and Rn concentrations in groundwaters, which are not new, and whose interest for the present study is not convincing.

I'm surprised that the discussion is no more hierarchical / structured around the following questions:

-What information can be deduced from the annual monitoring of the hydrogeochemical parameters analyzed before, during and after the earthquake, in terms of hydrogeological modifications of water reservoirs related to the earthquake.

- How the geochemical data collected after the seismic event, in particular the Sr and Rn data (but perhaps not only, because the other chemical information is not really discussed in the article) can be used to constrain the different scenarios based on the annual monitoring or to choose among them.

Also, I do not really understand the relevance of the SOM analysis, as made and used in this paper. I'm wondering if making the SOM analysis at the start of the article does not lead the authors to forget to do a relatively extensive presentation and discussion of the data, especially the geochemical data, relevant for their purpose. The latter is to build a sound conceptual model to explain possible mechanisms for the hydrological and geochemical responses of groundwaters to the earthquake.

The SOM analysis indicates the presence of strongly related parameters. Why not rely on this information to examine in more detail the key geochemical parameters, and to discuss their variability in binary or other diagrams, in order to evaluate their meanings in terms of water reservoirs, of water pathways,.., involved in the formation of groundwaters, and that could have been modified in response to the earthquake..?

Actually, very little is done with the geochemical data: just a rapid presentation of the data in the 87Sr/86Sr vs. 1/Sr and Ca vs Sr diagrams. Why? Is it because they do not help much? Why not looking at Piper diagrams for example or other binary mixing

diagrams, that can give information about the different sources potentially involved in the geochemical constitution of groundwaters (in terms of rock water interaction or in terms of water mixing)

To summarize, the construction of the paper under its present form is for me very confused. At this stage the interpretations remain very hypothetical and poorly justified by the data, even if the database is of good quality and the question of how to constrain the hydrological modifications related to earthquakes is interesting.

Therefore, I do not recommend publication of this manuscript under its present form: I encourage the authors to restructure and rewrite their paper in order to better justify and defend their interpretations, before resubmitting it at HESS or in another journal.

Other comments

L.253 and L. 358-359: If the SOM analysis simply leads to conclude that the classification obtained is close to the classification based on lithostratigraphic unit data, we can question the interest of such an analysis. It is well established today that at the first order the chemical composition of groundwater is controlled by the interactions of the waters with the aquifer rocks!

As already suggested above, would not it be more relevant to use some key geochemical parameters to evaluate if the geochemical differences between the different groups or the geochemical dispersion within a single water group can or cannot be related to hydrological characteristics of the aquifers (connectivity between reservoirs for example,). Such information could eventually be used as arguments to prove or defend some hypotheses made in the discussion section.

L. 345-356: The conclusion, that the large variation of Sr isotopic ratios in the groundwater can be explained by the nature of the aquifer lithology, is again not a very new conclusion.

L. 361-362 I do not understand why this grouping is different as the one given L 253

L. 372 : All what explained here is maybe right but without sound arguments it is difficult to be convinced/ Why invoking exchanges processes...based on which observation? If there is no sound observation, it is a possible scenario, but likely, one among others....

Idem L. 400 when is invoked a seawater intrusion...

L. 391- 392 : I fully agree with the authors that with only geochemical and isotopic data on water samples collected after the earthquakes, it is difficult here to be very conclusive ("it is difficult to confidently determine an effect of upwelling because data were only collected after the earthquake, not prior"). This is true here but more or less all along the discussion. It is why, above, I have suggested the authors to discuss first the annual monitoring data, the only one collected before, during and after the earthquake, and then only the other geochemical data, including Sr isotope ratios and Ra concentration data.

---

## Author Comment (AC2) · 7 Nov 2018

Dear Editor and Referee, Taking into account the useful comments we received from two referees, we provided a new version of the manuscript with adequate corrections. We already uploaded our responses to Referee #1's comments and details how we revised out manuscript. We enclose below our responses to Referee #2's comments, which surely improve the initial manuscript. We are confident that provided changes are sufficient for reconsidering our manuscript. Best Regards, For the Authors

Comments provided by Anonymous Referee #2

1) general comments The manuscript submitted for publication by Kim et al., in HESS seeks to evaluate the impact of seismic events on groundwater dynamics and geochemistry, in the case of the September 12, 2016 Gyeongju earthquakes in Korea. This scientific issue is of importance and in the scope of the scientific themes published in HESS. The work is based on an annual monitoring of groundwater level, temperature and electro-conductivity of several wells in the geographical area impacted by the seism. Such a monitoring should allow the authors to study the variations of the above parameters before, during and after the earthquakes, which should be at the heart of a discussion on the potential hydrological modifications linked to the seismic events, which is actually not the case. The authors rather based their discussion on geochemical data, including Sr isotope ratios and Rn data, of water samples collected in January 2017, after the seism and on a statistical analysis ("Self-Organizing Map (SOM) " ) of the hydrogeochemical characteristics of the groundwaters. Except if I have misunderstood the manuscript, I do not really understand such a choice, and I have many difficulties to really understand the arguments developed by the authors to sustain/defend the interpretations given in the discussion. The discussion under its present form is based on many general considerations on the origin of Sr isotope ratios and Rn concentrations in groundwaters, which are not new, and whose interest for the present study is not convincing. I'm surprised that the discussion is no more hierarchical / structured around the following questions: -What information can be deduced from the annual monitoring of the hydrogeochemical parameters analyzed before, during and after the earthquake, in terms of hydrogeological modifications of water reservoirs related to the earthquake. - How the geochemical data collected after the seismic event, in particular the Sr and Rn data (but perhaps not only, because the other chemical information is not really discussed in the article) can be used to constrain the different scenarios based on the annual monitoring or to choose among them. Also, I do not really understand the relevance of the SOM analysis, as made and used in this paper. I'm wondering if making the SOM analysis at the start of the article does not lead the authors to forget to do a relatively extensive presentation and discussion of the data, especially the geochemical data, relevant for their purpose. The latter is to build a sound conceptual model to explain possible mechanisms for the hy-

drological and geochemical responses of groundwaters to the earthquake. The SOM analysis indicates the presence of strongly related parameters. Why not rely on this information to examine in more detail the key geochemical parameters, and to discuss their variability in binary or other diagrams, in order to evaluate their meanings in terms of water reservoirs, of water pathways,.., involved in the formation of groundwaters, and that could have been modified in response to the earthquake..? Actually, very little is done with the geochemical data: just a rapid presentation of the data in the 87Sr/86Sr vs. 1/Sr and Ca vs Sr diagrams. Why? Is it because they do not help much? Why not looking at Piper diagrams for example or other binary mixing diagrams, that can give information about the different sources potentially involved in the geochemical constitution of groundwaters (in terms of rock water interaction or in terms of water mixing) To summarize, the construction of the paper under its present form is for me very confused. At this stage the interpretations remain very hypothetical and poorly justified by the data, even if the database is of good quality and the question of how to constrain the hydrological modifications related to earthquakes is interesting. Therefore, I do not recommend publication of this manuscript under its present form: I encourage the authors to restructure and rewrite their paper in order to better justify and defend their interpretations, before resubmitting it at HESS or in another journal.

RESPONSE: We thank the referee for taking his/her time to review our paper. We have attempted to satisfy all suggestions, so it made us to produce a stronger paper, adequate to be published on HESS. Please see the responses to the referee' comments below and subsequent changes in the revised manuscript (marked in red color). Thanks in advance for reconsidering the revised manuscript positively. If we understood correctly, Referee # 2 gave questions about the two most important issues in our paper. The two major issues are: 1) extensive interpretation with pre-, co-, and post-seismic monitoring data; and 2) most fundamental and important analyses based on hydrogeochemical data through traditional but most fundamental ways of analysis such as Piper diagram, binary mixing, and comparison of major components. We completely agree with the suggestions of the Referee #2 regarding what should have been done

at first with our monitoring data. The comments of the Referee #2 have pointed out precisely the problems that the authors have been troubled with. We have pre-, co-, and post-seismic time-series monitoring data on water level, temperature, and electric conductivity thanks to the operation of the national groundwater monitoring stations in Korea. However, they did not have the geochemical data of major groundwater constituents because few earthquakes usually occur in Korea. There have been quite many efforts by hydrogeologists to find or interpret any hydrogeologic changes by the earthquakes focused on the time-series monitoring data. However, regardless existence of the earthquake, water level and electrical conductivity showed ups and downs at scattered locations according to other factors such as seasonal effect. In addition to that, major chemical constituents only showed some noticeable difference between alluvial and bedrock aquifers in the Piper diagram, but not indicative of changes due to seismic events. This is why we tried to do more statistical clustering first and interpretation focused on the isotopes data for finding a mechanism, which explains the hydrogeologic responses to the earthquakes for each of the clusters derived. The comments of the Referee #2 might be based on the argument that a grouping or clustering that could be accomplished by a statistical analysis should also be possible by using major ions and few representative geochemical parameters. For example, such method like binary mixing models using major chemical constituents should be applied to wells showing mixing of deep geothermal waters. We agree with this point. What was done here in our study is to apply a statistical method first by using extended list (or most measured items) of hydrogeochemical data available for extracting similarity patterns, i.e., clustering. And then, each cluster is reasonably explained with hydrogeology. We hope it is well understood that the authors have chosen an approach for better grouping of the wells showing correlation with similar hydrogeological conditions. Regarding the time-series of water level, temperature, and electric conductivity data, temporal changes in sequence of pre- co- and post- seismic events usually do not have consistency in trends. There are many monitoring wells showing similar changes regardless of the seismic events. This is limitations of that data. Moreover, it was
difficult to derive a reasonable hydrologic interpretation with only the major ionic constituents because we do not have information on end member concentrations of the major constituents. The isotopic data (radon and strontium) employed in this study, on the other hands, made us possible to tell the groundwater in specific wells was affected by the earthquake. Based on the comments by the Referee #2, we also tried to reorganize our manuscript focused on the followed four points. The details of revision are as follows: First, we have divided the discussion section into three sub-chapters for more clear and logical interpretation and for better hierarchical/structured discussion as follows: 5.1 Groundwater level, temperature, and EC changes 5.2 Isotopic data (radon and strontium) 5.3 Conceptual model with the grouping results In addition, the results section has been also rearranged as follows: 4.1 Groundwater level, temperature, and EC changes 4.2 Hydrogeochemical characteristics including isotopes (radon and strontium) 4.3 Self-Organizing Map (SOM) Second, we have totally revised the discussion section to reflect the referee's main/minor comments. The hydrochemical data analysis has been included in section 4.2 (L258-264) and the Piper diagram has been added in section 5.3 to avoid one sided interpretation using the Sr and Rn data (Fig. S1). Before submission of our original paper, we had tried to draw and analyze the Piper diagram. However, the reason why we did not use the diagram in the original paper was that the diagram only considered major ions (Na, K, Ca, Mg, Cl, $SO_4$, and $HCO_3$) and it was difficult to find an explanatory basis without including other hydrochemical parameters ($NO_3$, Sr, $^{87}Sr/^{86}Sr$, temperature, pH, DO, EC, TDS, and salinity). The Piper diagram showed the distinct trend when the plot was drawn by dividing the groundwater wells into two groups; the alluvial aquifer wells and the bedrock aquifer wells. While the diagram indicated the ionic composition characteristics depended on the groups of water sample wells, it was not appropriate to estimate changes or differences due to the effects of the earthquakes. This diagram can also be used to explain the water-rock interactions in general, but a total system should be considered for interpreting the changes due to the earthquakes. In this regards, for more convincing interpretation about an overall characterization of the groundwater system

related to the earthquakes, we tried the SOM methods by using 16 hydrogeochemical parameters (Na, K, Ca, Mg, Cl, NO3, SO4, HCO3, Sr, 87Sr/86Sr, temperature, pH, DO, EC, TDS, and salinity). However, as the referee pointed out, we agree that the Piper diagram is also important for understanding water chemistry and quality, so we have added the diagram as Supplementary Figure 1. If it is recommended to be included in the main paper, we would add the Piper diagram as one of the main figures, not as a supplementary figure. The interpretation of 87Sr/86Sr vs. 1/Sr and Ca vs Sr diagrams has been also used in the discussion section (indicated see Fig.8). Third, the SOM analysis showed the correlation and clustering results graphically. The SOM method has an advantage in visualization of the multi-dimensional data, which is helpful to identify the dependencies between the variables (e.g. hydrogeochemical and isotopic data) and to classify the wells. This statistical method is not newly developed in this study, however, our study showed very interesting results that the grouping was in accordance with the lithostratigraphic units. This is not common case because many other results (in other sites or other time periods) did not show the correlation each other. Especially, this method also provided the detailed local relationship between the variables by the component planes, which was helpful to understand groundwater systems visually (L305-309). The local interpretation is important for the studies related to the earthquakes. In addition, as mentioned above, this method used the extended various hydrochemical parameters (Na, K, Ca, Mg, Cl, NO3, SO4, HCO3, Sr, 87Sr/86Sr, temperature, pH, DO, EC, TDS, and salinity). Based on the efficient explanatory ability of the SOM method for the groundwater study related to the earthquakes, we can suggest the application of SOM to researches in other sites for making statistically explanatory basis and then provide geological and hydrogeological interpretations of the observed phenomena. However, as the referee mentioned, we agree that the statistical results should be explained in close relations with the hydrogeochemical data and variables. Thus, we have entirely revised the discussion section. Please see the revised manuscript. Fourth, the groundwater level, temperature, and EC monitoring data before, during and after the earthquake and the geochemical data collected after

the seismic event were used in this paper. Because few earthquakes usually occur in Korea, especially in the study site, the data before the earthquake was insufficient. For this, we tried to add after-earthquake data by sampling most of national groundwater monitoring wells near the epicenters. We used additional data to deduce the relations between the origin of Sr isotope ratios and Rn concentrations in rocks and groundwater and the effects of the earthquakes. We think the proposed possible mechanisms of hydrogeological system changes due to the earthquakes in the study site are very important to extending scie3ntific understanding on the characteristically very local, heterogeneous, and irregular of the groundwater system to earthquakes.

2) Specific comments L.253 and L. 358-359: If the SOM analysis simply leads to conclude that the classification obtained is close to the classification based on lithostratigraphic unit data, we can question the interest of such an analysis. It is well established today that at the first order the chemical composition of groundwater is controlled by the interactions of the waters with the aquifer rocks! As already suggested above, would not it be more relevant to use some key geochemical parameters to evaluate if the geochemical differences between the different groups or the geochemical dispersion within a single water group can or cannot be related to hydrological characteristics of the aquifers (connectivity between reservoirs for example,). Such information could eventually be used as arguments to prove or defend some hypotheses made in the discussion section. RESPONSE: Thank you for comments. As the response to main comments above, to avoid one sided interpretation using the radon and strontium isotopic data, we have entirely revised the results and discussion section. Please see the revised manuscript. We have also added more interpretation of geochemical data including the correlations results of the SOM (Fig. 9) and 87Sr/86Sr vs. 1/Sr and Ca vs Sr diagram (Fig.6 and Fig. 7). In addition, as mentioned above responses, the SOM was conducted using geochemical dataset (Na, K, Ca, Mg, Cl, NO3, SO4, HCO3, Sr, 87Sr/86Sr, temperature, pH, DO, EC, TDS, and salinity), not including geological data. Some results did not show the high correlation between the SOM grouping results and the lithostratigraphic unit data in other researches and there are few cases using

SOM for the hydrological systems related to the earthquakes. Moreover, this method is useful for understanding groundwater systems visually by 2D diagram. Thus, we suggested that this is also helpful for analyzing hydrochemical characteristics, not as the only method.

L. 345-356: The conclusion, that the large variation of Sr isotopic ratios in the ground-water can be explained by the nature of the aquifer lithology, is again not a very new conclusion. RESPONSE: We agree with your comments. For the background information, the general values and ranges were written in that part. As mentioned above responses, the data was insufficient in the study site because few earthquakes occur in Korea. It is the process for further study.

L. 361-362 I do not understand why this grouping is different as the one given L 253 RESPONSE: In L322, the grouping was conducted as follows: Group 1 (KW 1, KW 2, KW 9-1, and KW 10-1), Group 2 (KW 3, KW 5-1, KW 5-2, KW 6-2, KW 11-3, and KW 12-1), Group 3 (KW 4-1 and KW 4-2), and Group 4 (KW 8-1, KW 11-1, and KW 11-2). The SOM does not include all input wells. The U-matrix shows the selected wells which showed high correlation each other. Thus, the results may not include some input wells. Our study results also did not include KW 6-1, KW 7-1, KW 7-2, KW 8-2, KW 9-2, KW 10-2, and KW 12-2 in the U-matrix. However, the classification results had high similarity with the classification based on lithostratigraphic unit data. In L175, the study area was divided into four sections; (i) Hayang-group shale and sandstone (KW 1, KW 2, KW 9-2, KW 10-2), (ii) Bulguksa-group biotite granite (KW 3, KW 5-2, KW 12-2), (iii) tuff and tuffaceous sedimentary rocks of Yeonil-group and Janggi group (KW 4-2, KW 6-2, KW 7-2), and (iv) Cretaceous volcanic rocks mainly composed of andesite (KW 8-2, KW 11-2). This lithostratigraphic unit data can be used for arranging the bedrock aquifer wells based on bedrock characteristics, so this classification does not include the alluvial aquifer wells. For convincing interpretation of this study, we conducted the new grouping to reflect the two grouping results (L175 and L253). In conclusion, the final grouping has been conducted combining L175 (lithostratigraphic
unit) and L322 (SOM results). This has been performed by binding the alluvial and bedrock aquifer wells: Group A (KW 1, KW 2, KW 9, and KW 10); Group B (KW 3, KW 5, and KW 12); Group C (KW 4, KW 6, and KW 7); and Group D (KW 8 and KW 11) (as written in L361). However, as the referee pointed out, this explanation was not sufficient in the manuscript, so we have written additional explanation (L380-400).

L. 372: All what explained here is maybe right but without sound arguments it is difficult to be convinced/ Why invoking exchanges processes. . .based on which observation? If there is no sound observation, it is a possible scenario, but likely, one among others.... RESPONSE: As suggested in general comment, we entirely revised the results and discussion section including additional hydrogeochemistry interpretation. Among them, Group A scenario was described focused on the Ca vs Sr diagrams and the SOM component maps. This group had high values and high positive correlations between Sr and Ca, which indicate the similar behavior in both rock and groundwater. This chemical parameter showed the one possible scenario, which is the strong water-rock interaction. This was illustrated in L419-421 and L462-464.

L. 400: when is invoked a seawater intrusion. . . RESPONSE: By considering the comments of the referee, the results and discussion section were revised entirely. Especially, the Group C mechanism, which suggested the possibility of sea water intrusion in the wells, was revised including the Piper diagram analysis (L437-442).

L. 391- 392 : I fully agree with the authors that with only geochemical and isotopic data on water samples collected after the earthquakes, it is difficult here to be very conclusive ("it is difficult to confidently determine an effect of upwelling because data were only collected after the earthquake, not prior"). This is true here but more or less all along the discussion. It is why, above, I have suggested the authors to discuss first the annual monitoring data, the only one collected before, during and after the earthquake, and then only the other geochemical data, including Sr isotope ratios and Ra concentration data. RESPONSE: Thank for considerable comments. As the response to main comments above, we have rearranged the contents of the paper; 4.

Results 4.1 Groundwater level, temperature, and EC changes 4.2 Hydrogeochemical characteristic including isotopes (radon and strontium) 4.3 Self-Organizing Map (SOM) 5. Discussion 5.1 Groundwater level, temperature, and EC changes 5.2 Isotopic data (radon and strontium) 5.3 The conceptual model with the grouping results We also have redrawn the Fig. 1 in detail. Please see the revised manuscript.

Please also note the supplement to this comment:
https://www.hydrol-earth-syst-sci-discuss.net/hess-2018-360/hess-2018-360-AC2-supplement.pdf
* * *
[Figure]

Group A
Group B
Group C
Group D

Jan., 2017

KW 12-2
KW 11-1
KW 1   KW 10-2
KW 9-1
KW 11-2   KW 12-1   KW 5-1
KW 4-2
KW 10-1
KW 8-2   KW 8-1
KW 9-2
KW 6-2
KW 7-2
KW 7-1
KW 2   KW 3
KW 4-1
KW 6-1
KW 5-2

$SO_4^{2-}+Cl^-$
$Ca^{2+}+Mg^{2+}$
$Mg^{2+}$
$SO_4^{2-}$
$Na^++K^+$
carbonate
$Ca^{2+}$
$Na^++K^+$
$HCO_3^-$
$Cl^-$

**Fig. 1.** Fig. S1.

---

## Author Comment (AC3) · 7 Nov 2018

**1 Hydrogeological responses to the 2016 Gyeongju earthquakes,**

**2 Korea**

- 3 Jaeyeon Kim1, Jungjin Lee1, Marco Petitta2, Heejung Kim1, Dugin Kaown1, In-Woo Park1,
- 4 Sanghoon Lee1 and Kang-Kun Lee1\*
- 1 School of Earth and Environmental Sciences, Seoul National University, Seoul 08826, Republic of Korea.
- 6 2Department of Earth Sciences, Sapienza University of Rome, P.le A. Moro 5, 00185 Rome, Italy.
- 7 *Correspondence to*: Kang-Kun Lee1\* (kklee@snu.ac.kr)

**8 Abstract**

9 The September 12, 2016 Gyeongju earthquakes ( $M_L = 5.1$  of foreshock and  $M_L = 5.8$  of mainshock) had significant effects on groundwater systems along the Yangsan Fault System 10 11 composed of NNE-trending, right-lateral strike-slip faults in Korea. Hydrological changes 12 induced by the earthquakes are important because no surface ruptures have been reported and few earthquakes usually occur in Korea. The main objective of this research was to propose a 13 conceptual model interpreting the possible mechanisms of groundwater response to the 14 earthquakes based on anomalous hydrogeochemical data including isotope concentrations 15 with lithostratigraphic classification. For this, annual monitoring data (groundwater level, 16 temperature, and electrical conductivity) and collected data (hydrochemical parameters, 17 radon-222, and strontium isotopes) were used. Groundwater level anomalies could be 18 attributed to the movement of the epicentral strike-slip fault. Radon concentration data 19 showed the potential of groundwater mixing processes. Strontium anomalies could be related 20 to the lithology and stratigraphy of the bedrock, reflecting the effect of water-rock interaction. 21 Using a Self-Organizing Map (SOM) statistical analysis, associations of hydro-geochemical 22 characteristics among groundwater wells were interpreted. By combining the grouped results 23 of the SOM with lithostratigraphic unit data, 21 groundwater wells were classified into four 24 groups, each corresponding to different hydrogeological behaviors. A new comprehensive 25 26 conceptual model was developed to explain possible mechanisms for the hydrological and 27 geochemical responses in each group, which have been respectively identified as water-rock

interaction, mixing of shallow and deep aquifers via sea water intrusion, bedrock fracture
opening related to strike-slip fault movement, and no response.

33

**1. Introduction**

Earthquakes have a great influence on groundwater hydrology, such as water table changes 34 and hydrochemical anomalies. Typically, most studies have focused on earthquake 35 forecasting, i.e. changes prior to earthquakes, or co-seismic behavior. There have been a 36 37 limited number of studies that discuss the responses of groundwater systems, especially, focused on the hydrogeologic changes after earthquakes (Adinolfi Falcone et al., 38 2012; Amoruso et al., 2011; Barberio et al., 2017; Claesson et al., 2007; Ekemen Keskin, 39 2010;Galassi et al., 2014;Lee et al., 2013;Matsumoto et al., 2003;Petitta et al., 2018;Wang 40 and Manga, 2010; Wang et al., 2012; Yechieli and Bein, 2002). Seismic waves, for example, 41 42 are known to cause changes in water level, temperature, and geochemistry (Matsumoto et al., 43 2003;Roeloffs et al., 2003;Roeloffs, 1998;Shi et al., 2015;Adinolfi Falcone et al., 2012;Wang et al., 2012). Seismicity can also cause abrupt changes or have long-term effects on the 44 environment, particularly, groundwater systems. Hydrological responses to seismicity depend 45 on several factors such as the earthquake magnitude, distance from the epicenter, the 46 chemical and physical properties of the water, geological structures, permeability, and the 47 pore pressure of rocks (Ekemen Keskin, 2010;Hartmann and Levy, 2005;Petitta et al., 2018). 48 For example, Ekemen Keskin (2010) stated that the observed changes in aquifers could be 49 explained using a dilatancy-fluid model; the response to earthquakes could be attributed to 50 51 the changes in the water mixing ratio because of aquifer permeability, pore pressure, and flow 52 path. Moreover, locally heterogeneous responses of groundwater have been observed and

53 associated with the dominant lithology and mineralogy of bedrocks (Frape et al., 1984;Shand 54 et al., 2009;Kim et al., 1996), local degree of deformation (Fitz-Diaz et al., 2011), or fracture 55 networks allowing groundwater flow (Gray et al., 1991). Some studies also have proposed 56 some conceptual models for describing the aquifer responses to earthquakes by analyzing hydraulic properties (Adinolfi Falcone et al., 2012; Amoruso and Crescentini, 2010; Amoruso 57 et al., 2011;Barberio et al., 2017;Manga, 2001;Roeloffs, 1998;Tokunaga, 1999). 58 On September 12, 2016, two earthquakes ( $M_L = 5.1$  and  $M_L = 5.8$ , respectively) occurred in 59 Gyeongju, in the southeastern part of the Korean Peninsula (Korea Meteorological 60 Administration). The mainshock of the Gyeongju events was recorded as the largest 61 62 earthquake in Korea since instrumental seismic monitoring started in Korea in 1978. The 63 source mechanism of the Gyeongju earthquakes displayed strike-slip movement of a branch of the Yangsan Fault (YSF) passing through the Gyeongju area (Kim et al., 2017b;Kim et al., 64 2016). Slip analysis and earthquake focal mechanism solutions have shown that the YSF is 65 under a regional compressional stress field which might be a result of the continental 66 collision of the Pacific, Eurasian, and Indian plates (Jiang et al., 2016;Park et al., 2007;Park 67 et al., 2006;Zoback, 1992). Gyeongju area is spatially close to the YSF, thus the Gyeongju 68 events might allow more detailed studies about characteristics of the YSF and its branch 69 70 faults, including groundwater responses after earthquakes.

71 The earthquake-related indicators in hydrogeology generally include (i) groundwater level,

5

(ii) temperature, (iii) hydrochemistry, and (iv) isotope concentrations. Groundwater level
monitoring has been broadly used to identify pre-, co-, and post- earthquake changes (BenZion and Aki, 1990;Brodsky, 2003;Manga et al., 2012;Roeloffs, 1998;Shi et al., 2015;Wang
and Manga, 2010). Seismic waves have been known to affect the groundwater level via
oscillations and permanent offsets. Temperature changes are commonly analyzed using heat
transport modeling (Ekemen Keskin, 2010;Wang et al., 2012).

In this study, groundwater chemistry, major elements, and some physical-chemical 78 parameters (pH, EC, and temperature) were monitored. Among the isotopes, oxygen, 79 hydrogen, and radon-222 were analyzed to determine the effects of the earthquake on 80 81 groundwater. Radon-222, particularly, has been generally monitored as an earthquake 82 precursor sampled in water or air (Igarashi et al., 1995;King, 1978;Liu et al., 1984;Noguchi and Wakita, 1977; Roeloffs, 1999; Teng, 1980; Wakita et al., 1980). Radon-222 is a radioactive 83 nuclide with a half-life of approximately 3.8 days. It is produced from radium-226 in the 84 natural radioactive decay chain of uranium-228; thus, its concentration is proportional to the 85 uranium concentration in adjacent rocks. The transport of radon is influenced by fluid 86 advection, diffusion, partition between the liquid and gas phases, and radioactive decay. The 87 radon concentration in groundwater is dependent on the specific surface of the rocks (Hoehn 88 and Von Gunten, 1989; Torgersen et al., 1990). Because the specific surface can be affected by 89 earthquakes, the radon concentration can increase or decrease. Radon-222 also shows 90 91 significant anomalies at fault zones prior to earthquakes (Ghosh et al., 2009; Walia et al.,

| 92  | 2009; Wang and Fialko, 2015). However, a few studies have delineated the response of radon                           |
|-----|----------------------------------------------------------------------------------------------------------------------|
| 93  | concentration to earthquakes (Adinolfi Falcone et al., 2012;Igarashi et al., 1993;Igarashi and                       |
| 94  | Wakita, 1990). Strontium isotopes have been used in only a few earthquake-related papers.                            |
| 95  | Strontium isotopes are useful tracers for groundwater origin and water-rock mixing processes                         |
| 96  | because ${}^{87}$ Sr is the daughter product of the natural decay of radioactive ${}^{87}$ Rb (half-life = 48.9      |
| 97  | Ga). The 87 Sr/ 86 Sr ratios of the bedrock aquifers were different according to rock types of |
| 98  | bedrock (Frape et al., 1984; Frost and Toner, 2004; Négrel et al., 2004; Shand et al., 2009).                        |
| 99  | Thus, strontium isotopes in groundwater can also reveal significant post-earthquake                                  |
| 100 | anomalies at fault zones according to bedrock type.                                                                  |

The main objective of this study was to identify hydrogeochemical changes related to the 101 102 Gyeongju earthquake and then suggest a conceptual model of the response of the groundwater systems to the earthquake using the grouping results. In accordance with this 103 objective, major research results were achieved via (i) performing a correlation and cluster 104 analysis of hydrochemical parameters with geological characteristics using the SOM 105 approach; (ii) analyzing pre-, co-, and post-seismic changes in groundwater level, 106 107 temperature, and EC; and (iii) interpreting the results of isotopes (radon and strontium) sampled following the earthquake based on the grouping. These results could help to provide 108 the possible mechanisms of groundwater changes induced by the earthquake. The overview 109 of this research is shown in Fig. 1. 110

**112 **2.** Study area**

The Gyeongju earthquake sequence started with a foreshock ( $M_L = 5.1$ ) at 10:44:32 UTC, on September 12, 2016, and the mainshock ( $M_L = 5.8$ ) occurred at 11:32:55 UTC (Korea Meteorological Administration). During the first 10 days following the mainshock, more than 120 earthquakes of  $M_L \ge 2.0$  were recorded in the epicentral region. The earthquakes,

including the mainshock and strong aftershocks ( $M_L \ge 3.5$ ) are listed in Table 1.

The Korean Peninsula is composed of three major Precambrian massifs: the Nangrim, 118 Gyeonggi, and Youngnam from north to the south. The Gyeongsang Basin is the northern 119 120 part of the Youngnam massif and the Yangsan Fault System has developed in the eastern part of the Gyeongsang Basin. The Yangsan Fault System is a group of NNE-trending major 121 strike-slip faults. The Gyeongju earthquake and its abundant aftershocks occurred near the 122 123 YSF (Fig. 2a), which has a linear expression for approximately 200 km, and is the longest major fault of the Yangsan Fault System (Kyung and Lee, 2006). The displacement of the 124 fault varies between 21 km and 35 km depending on the location, and the arrangement of the 125 granitic rocks in this area indicates Cenozoic dextral strike-slip of 21.3 km in the N 20° E 126 direction along the YSF line (Hwang et al., 2004;Hwang et al., 2007). 127

128 Since Lee and Na (1983) first suggested that a Quaternary reactivation of the Yangsan Fault System could be possible, a number of seismic, geological, and geophysical studies have 129 130 proved its seismic activation (Kyung and Lee, 1999;Kyung and Lee, 2006;Lee and Jin, 1991). 131 The Gyeongju area has been subject to most of the large historical earthquakes that have occurred in Korea. Initial movement on the YSF was recorded to have occurred before 45 132 Ma, based on radiometric dating of volcanic rocks (Chang et al., 1990). Age dating using the 133 accelerator mass spectrometry (AMS) method indicates late Quaternary movement of the 134 YSF between 2,400 and 2,000 yrs BP and an average vertical slip rate of approximately 0.04-135 136 0.05 mm/yr (Kyung and Chang, 2001). Recent measurement of the vertical slip rate of YSF reported less than 0.1 mm/yr on average (0.02–0.07 mm/yr in the southern part and 0.03–0.05 137 mm/yr in the northern part) (Kyung, 2003;Kyung and Lee, 2006), indicating that the YSF has 138 been seismically active. 139

140 Paleo-stress analyses have noted that the stress regime of the YSF has changed more than 141 three times (Kang and Ryoo, 2009;Kim et al., 1996), and during the Quaternary the ENE-WSW maximum compression is in agreement with the first-order stress field in east Asia 142 143 (Chang et al., 2010;Heidbach et al., 2010;Zoback, 1992). Trench analysis of the Yugye Fault, the youngest Quaternary fault in the northern part of the YSF, also yielded a NW-SE or 144 145 WNW-ENE compressional local maximum principal stress (Kim and Jin, 2006). For the Gyeongju event, also under this ENE-WNW compression, geophysical studies of the 146 aftershocks recognized the subsurface fault plane has a strike of NNE 25-30° and a dip of 147

[revised manuscript text omitted]

**4.2 Hydrogeochemical characteristics including isotopes (radon and strontium)**

The hydrogeochemical data of 17 parameters (Na, K, Ca, Mg, Cl, NO3, SO4, HCO3, temperature, pH, DO, EC, TDS, salinity, Sr, 87Sr/86Sr, and radon) were collected from 21 groundwater samples and one surface water sample (KW 11-3) in January 2017. The analytical results of the water samples are summarized in Table 3. The Na values were high in KW 4-1, KW 4-2, and KW 6-2 (> 100 mg/L), and the Cl values were high in KW 4-1 and KW 4-2 (> 100 mg/L). The KW 4-1 and KW 4-2 had especially high EC and salinity values.

| 265 | The distribution of radon concentration in the 21 groundwater samples is shown in Fig. 5.           |
|-----|-----------------------------------------------------------------------------------------------------|
| 266 | The radon concentration ranged from 225 $Bq/m^3$ to 23060 $Bq/m^3$ in the Gyeongju area (see        |
| 267 | Table 3). The KW 5-1, KW 5-2, and KW 8-2 values were 15849, 17575, and 23060 $Bq/m^3$ ,             |
| 268 | respectively, which were higher values than those of the other groundwater wells. These wells       |
| 269 | are near the epicenter. Lower values (< $1000 \text{ Bq/m}^3$ ) were found in KW6-1, KW 6-2, KW 7-1 |
| 270 | KW 9-1, KW 9-2, KW 10-1, KW 10-2, and KW 11-1. The values between the alluvial and                  |
| 271 | bedrock aquifer wells were similar in KW 4, KW 5, KW 6, and KW 7. The value difference              |
| 272 | between two formation wells was high in KW 8, KW 10, and KW 11. KW 7, KW 9, and KW                  |
| 273 | 11 showed an anomaly in which the alluvial aquifer well had a higher radon concentration            |
| 274 | than the bedrock aquifer well.                                                                      |

The strontium isotopic compositions of groundwater samples in the Gyeongju area are 275 shown in Fig. 6. Strontium concentrations ranged from 18.1 ppb to 4052 ppb. The 87Sr/86Sr 276 values ranged from 0.705688 to 0.712368 (see Table 3). In the alluvial aquifer wells, the 277  $^{87}$ Sr/ $^{86}$ Sr values ranged from 0.706191 to 0.708353, and these values were from 0.705688 to 278 0.712368 in the bedrock aquifer wells. The strontium isotopic compositions of the 279 groundwater samples also reflected distinct ratios based on their lithology and stratigraphy. 280 The Hayang group (KW 1, KW 2, KW 9-2 and KW 10-2) had high strontium concentrations. 281 The 87Sr/86Sr values of the Bulguksa group (KW 3, KW 5-2, and KW 12-2) ranged from 282

[revised manuscript text omitted]

**352 **5.2 Isotopic data (radon and strontium)**

The hydrogeochemical data of 17 parameters (Na, K, Ca, Mg, Cl, NO3, SO4, HCO3, temperature, pH, DO, EC, TDS, salinity, Sr, 87Sr/86Sr, and radon) were collected only after earthquake (Jan., 2017). A difference in radon concentration between the alluvial and bedrock aquifer wells could be considered more significant because of the mixing effect as observed in KW 8 and KW 11 (see Fig. 5). Seismotectonic activity may often change the mixing ratio of groundwater in a well (Claesson et al., 2007;Hartmann and Levy, 2005). The anomaly in which the alluvial aquifer well had a higher radon concentration than that of the bedrock 360

361

aquifer could be attributed to rainfall; however, in this area, rainfall did not occur during the sampling period (as observed in KW 7, KW 9, and KW 11).

The large variation in the 87Sr/86Sr ration in the groundwater can consequently largely be 362 explained by the nature of the aquifer lithology (see Fig. 6 and Fig. 7). For example, the high 363 Rb/Sr ratio of composite silicate minerals such as feldspar and biotite can cause granitic 364 bedrock to be highly radiogenic (Frost and Toner, 2004;Santoni et al., 2016). Generally, 365 Cretaceous granites comprising the Gyeongsang basin had a strontium concentration from 62 366 ppm to 428 ppm and an  ${}^{87}$ Sr/ ${}^{86}$ Sr ratio from 0.704610 to 0.711400 (Cheong and Jo, 2017). 367 Basaltic rocks near the Yeonil group and Janggi group had strontium concentration from 439 368 ppm to 518 ppm and the 87Sr/86Sr ratio from 0.703850 to 0.704630 (Shimazu et al., 1990). In 369 the Chaeyaksan basaltic volcanics of the Yucheon group, strontium concentration ranged 370 from 731 ppm to 1667 ppm and the 87Sr/86Sr ratio from 0.705870 to 0.706440 (Yun, 1998). 371 Thus, samples of the Bulguksa granite would be more radiogenic than those of the Yucheon 372 group rocks, because of the composite minerals of the Bulguksa granite (plagioclase, feldspar, 373 and biotite) which have high 87Sr/86Sr ratios. The strong correlation between strontium and 374 375 calcium could indicate the chemical signature such as water-rock interactions (see Fig. 8). The outlier, KW 2, KW 4-2, and KW 11-2, can indicate another chemical mechanism such as 376 other water source. The high values of KW 9-1 and KW 10-1 can suggest the influence from 377 the KW 9-2 and KW 10-2. 378

**379 5.3** The conceptual model with the grouping results**

This paper is to combine the hydrologic, hydrogeochemical, and lithostratigraphic 380 characteristics by applying the neural network method SOM. The SOM results using the 381 hydrogeochemical data showed the 4 groups, however, these did not include KW 6-1, KW 7-382 1, KW 7-2, KW 8-2, KW 9-2, KW 10-2, and KW 12-2 in the U-matrix. For including all 383 groundwater samples, the new grouping was conducted to reflect the two grouping results 384 (lithostratigraphic unit and the SOM). The SOM results were also in agreement with the 385 lithostratigraphic unit data, which was used usefully in arranging the bedrock aquifer wells 386 based on bedrock characteristics. Especially, this statistical method could consider the factor 387 388 of external effects, e.g. earthquakes, as well as the major ions. It is useful because the public activity generally has been existed around the groundwater wells. Among other methods, in 389 general, Piper diagram provide relative proportions of water-rock interactions, which is 390 difficult to indicate directly the external effects other than geologic characteristics. Thus, by 391 using the two results, the final grouping yielded four classes of wells: Group A (KW 1, KW 2, 392 KW 9, and KW 10); Group B (KW 3, KW 5, and KW 12); Group C (KW 4, KW 6, and KW 393 7); and Group D (KW 8 and KW 11). This is conducted by binding the alluvial and bedrock 394 aquifer wells. The dependencies between the variables, hydrochemical parameters including 395 strontium isotopes, could be interpreted with the component plane results from the SOM (see 396 Fig. 9). These correlations were also used for analyzing the possible mechanisms at each 397 398 group. The piper diagram was also analyzed with the groups (Fig. S1). In this diagram, most water samples were prevailing SO4 and HCO3, suggesting the possibility of water body
mixing with other water type.

| 401 | The lithology and stratigraphy of Group A is classified as Hayang group shale and sandstone                          |
|-----|----------------------------------------------------------------------------------------------------------------------|
| 402 | of low porosity and high strontium concentrations. Particularly, the KW 9 and the KW 10                              |
| 403 | wells had a low radon concentration (< 1000 Bq/m3), high strontium concentration, high Ca                            |
| 404 | value, low 87 Sr/ 86 Sr ratio and low pH (see Fig. 8 and Fig. 9). There might be some possible |
| 405 | mechanisms for the exceptionally strong chemical signatures. Regarding earthquakes, first,                           |
| 406 | the fine-grained bedrock of Group A has a large reactive surface area that can effectively                           |
| 407 | activate water-rock interaction and largely vary the groundwater chemistry via ion exchange                          |
| 408 | (Pennisi et al., 2006). Second, particularly for KW 10-2 in Group A, the exceptionally high Sr                       |
| 409 | samples appear to be an effect of cation exchange between the soil and surrounding water.                            |
| 410 | The capacity of the cation-bearing soil (cation-exchange capacity; CEC) depends on the pH                            |
| 411 | of the surrounding water, and the CEC of a soil generally show decreases with pH decreases                           |
| 412 | (Sparks, 2003). The acidic water of KW10-2 ( $pH = 2.27$ ) would lead to lower CEC and lead                          |
| 413 | to leaching of $Ca^{2+}$ and $Sr^{2+}$ from the soil to surrounding groundwater. The flow into the                   |
| 414 | groundwater in the Hayang group rocks could increase the chemical concentration of Group                             |
| 415 | A. Third, the results could be attributed to geological characteristics, not related to the                          |
| 416 | earthquake, as the intrinsic chemistry of the Hayang group shale and sandstone might affect                          |
| 417 | the strontium concentrations. Such dramatically high values of Sr were previously observed                           |
| 418 | in the Redbeds aquifer (from 885 ppb to 7851 ppb) where the lithology of the bedrock is                              |

composed of shale and sandstone with high Rb/Sr ratios (Santoni et al., 2016). Moreover, by
the SOM results, KW 1, KW 2, KW 9-1, and KW 10-1 were clustered as one group (see Fig.
10), also suggesting the strong influence from bedrock to shallow aquifer.

Group B wells are located in biotite granitic region of the Bulguksa group, which has a 422 typical high radon concentration. The radon concentration is greatly influenced by uranium 423 content; thus, its concentration is generally high in granite compared to that of sedimentary 424 rocks. Typically, uranium concentration is high in granites, whereas it is low in sedimentary 425 rocks. However, only the KW 5 wells had a high radon concentration. In particular, KW 5-1 426 had high values similar to those of KW 5-2. This could be attributed to deep fluid upwelling 427 from the bedrock in the KW 5 wells (Chiodini et al., 2000; Minissale, 2004; Savoy et al., 2011). 428 In addition, the KW 5-1 was located toward  $SO_4^{2+}+CI^{-}$  in the piper diagram (see Fig. S1), 429 which indicates the deep groundwater (Reddy and Nagabhushanam, 2012). 430

Group C is composed of tuff and tuffaceous sedimentary rocks of the Yeonil and Janggi groups. This group had a low radon concentration and a small difference in radon concentration between the alluvial and the bedrock aquifer wells (see Fig. 5), suggesting active water mixing between the two aquifers. In addition, the bedrock of this area contains conglomerates, which generally have high pore density, leading to active mixing with water compared to the shale-dominant lithology. This hypothesis seems to be consistent with the weak chemical signature of Group A. KW 4-1, KW 4-2, and KW 6-2 had high values of EC, Cl, TDS, and salinity values (see Table 3 and Fig. 9), suggesting the possibility of sea water
intrusion by the effects of the earthquakes. The piper diagram also indicated the sea water
intrusion by the location of Group C, which got toward Na++K+ and HCO3- (see Fig. S1).
These water samples were prevailing HCO3 water. Sea water intrusion might actively trigger
mixing between the shallow and deep aquifers.

In Group D, the radon concentration was quite different between the two wells and the 443 groundwater level anomaly occurred (see Fig. 3 and Fig. 5). The wells of this group are in 444 Cretaceous mainly andesitic volcanic rocks. This group showed obvious anomaly caused by 445 the earthquakes with the annual data. The maintenance of a groundwater level increase was 446 447 observed in KW 8 wells, indicating the possibility of aquifer compaction. KW 11-1, which showed the greater decrease of groundwater level prior to the earthquake, suggested the 448 possibility of the opening of bedrock fractures. In addition, KW 11 wells, in particular, 449 showed many factors including groundwater level, temperature, and EC responded to the 450 earthquake in an opposite manner (see Fig. 3 and Fig. 4). The radon concentration of KW 11-451 1 was also higher than that of KW 11-2. The Sr contents of Group B and Group D show a 452 wide range of concentrations observed in other studies of groundwater in the granitic bedrock 453 aquifers; e.g., an Sr2+ from 67 to 169 ppb (Frost and Toner, 2004) and from 103 to 553 ppb 454 (Santoni et al., 2016). This wide range might be associated with the different amount of 455 plagioclase in each matrix rock of the groundwater. Water flow via granite can be controlled 456 457 by the dissolution of anorthite and alkali feldspar. The former occurs more rapidly, providing

Ca2+ and Sr2+ with a low 87Sr/86Sr ratio (see Fig. 8 and Fig. 9) (Bullen et al., 1997;Franklyn et al., 1991;Négrel, 2006). In contrast, one groundwater chemistry study in Canada showed that dissolution of alkali feldspar can increase the 87Sr/86Sr ratio providing sodium and potassium (Bullen et al., 1996). Therefore, the various compositions of the granite and the fluid mobility would be determinative in the 87Sr/86Sr ratio. Moreover, KW 11-1 also located in high SO42- +CI° than KW 11-2 in the piper diagram, suggesting the deep water influence to KW 11-1 (Reddy and Nagabhushanam, 2012).

[revised manuscript text omitted]

- 650 Tectonophysics, 180, 237-254, 1990.
- Igarashi, G., Tohjima, Y., and Wakita, H.: Time-variable response characteristics of
  groundwater radon to earthquakes, Geophysical research letters, 20, 1807-1810, 1993.
- Igarashi, G., Saeki, S., Takahata, N., Sumikawa, K., Tasaka, S., Sasaki, Y., Takahashi, M., and
- Sano, Y.: Ground-water radon anomaly before the Kobe earthquake in Japan, Science, 269,60-61, 1995.
- Jiang, L., Qiu, Z., Wang, Q., Guo, Y., Wu, C., Wu, Z., and Xue, Z.: Joint development and
- tectonic stress field evolution in the southeastern Mesozoic Ordos Basin, west part of North
- 658 China, Journal of Asian Earth Sciences, 127, 47-62, 2016.
- 659 Kang, J.-H., and Ryoo, C.-R.: The movement history of the southern part of the Yangsan
- 660 Fault Zone interpreted from the geometric and kinematic characteristics of the Sinheung Fault,
- Eonyang, Gyeongsang Basin, Korea, The Journal of the Petrological Society of Korea, 18,19-30, 2009.
- 663 Kim, K. H., Kim, J., Han, M., Kang, S. Y., Son, M., Kang, T. S., Rhie, J., Kim, Y., Park, Y.,
- and Kim, H. J.: Deep Fault Plane Revealed by High-Precision Locations of Early Aftershocks
- Following the 12 September 2016 ML 5.8 Gyeongju, Korea, Earthquake, Bulletin of the
- 666 Seismological Society of America, 108, 517-523, 2017a.
- 667 Kim, Y., Jang, B.-A., and Park, S.: Open microcracks in granites from the Yangsan fault zone
- and the stress field of the Kyeongsang Basin, Journal of the Geological Society of Korea, 32,367-378, 1996.
- 670 Kim, Y., and Jin, K.: Estimated earthquake magnitude from the Yugye Fault displacement on
- a trench section in Pohang, SE Korea, Journal of the Geological Society of Korea, 42, 79-94,
  2006.
- Kim, Y., Rhie, J., Kang, T.-S., Kim, K.-H., Kim, M., and Lee, S.-J.: The 12 September 2016
- Gyeongju earthquakes: 1. Observation and remaining questions, Geosciences Journal, 20,
  747-752, 2016.
- Kim, Y., He, X., Ni, S., Lim, H., and Park, S. C.: Earthquake Source Mechanism and Rupture
- Directivity of the 12 September 2016 M w 5.5 Gyeongju, South Korea, Earthquake, Bulletin
- of the Seismological Society of America, 107, 2525-2531, 2017b.
- King, C.-Y.: Radon emanation on San Andreas fault, Nature, 271, 516, 1978.

- King, G., and Cocco, M.: Fault interaction by elastic stress changes: New clues from
  earthquake sequences, in: Advances in Geophysics, Elsevier, 1-VIII, 2001.
- Kitagawa, Y., Koizumi, N., Takahashi, M., Matsumoto, N., and Sato, T.: Changes in
  groundwater levels or pressures associated with the 2004 earthquake off the west coast of
  northern Sumatra (M9. 0), Earth, planets and space, 58, 173-179, 2006.
- Kohonen, T.: Self-organized formation of topologically correct feature maps, Biologicalcybernetics, 43, 59-69, 1982.
- Kyung, J., and Chang, T.: The latest fault movement on the northern Yangsan fault zone
  around the Yugye-ri area, southeast Korea, Journal of the Geological Society of Korea, 37,
  563-577, 2001.
- 690 Kyung, J. B., and Lee, G. H.: A paleoseismological study of the Yangsan fault-analysis of
- deformed topography and trench survey, Journal of the Korean Geophysical Society, 2, 1999.
- Kyung, J. B.: Paleoseismology of the Yangsan fault, southeastern part of the Korean
  peninsula, Annals of Geophysics, 46, 983-996, 2003.
- 694 Kyung, J. B., and Lee, K.: Active fault study of the Yangsan fault system and Ulsan fault
- system, Southeastern part of the Korean Peninsula, Journal of Korean Geophysical Society, 9,219-230, 2006.
- Lee, J., Ryoo, Y., Park, S. C., Ham, Y. M., Park, J. S., Kim, M. S., Park, S. M., Cho, H. G.,
  Lee, K. S., and Kim, I. S.: Seismicity of the 2016 ML 5.8 Gyeongju earthquake and
  aftershocks in South Korea, Geosciences Journal, 22, 1-12, 2018.
- Lee, K., and Na, S. H.: A study of microearthquake activity of the Yangsan fault, Journal of
  the Geological Society of Korea, 19, 127-135, 1983.
- Lee, K., and Jin, Y. G.: Segmentation of the Yangsan fault system: geophysical studies on
  major faults in the Kyeongsang basin, Journal of the Geological Society of Korea, 27, 434449, 1991.
- Lee, M., Liu, T.-K., Ma, K.-F., and Chang, Y.-M.: Coseismic hydrological changes associated
- with dislocation of the September 21, 1999 Chichi earthquake, Taiwan, Geophysical
  Research Letters, 29, 5-1-5-4, 10.1029/2002gl015116, 2002.
- Lee, S.-H., Ha, K., Hamm, S.-Y., and Ko, K.-S.: Groundwater responses to the 2011 Tohoku
- Earthquake on Jeju Island, Korea, Hydrological Processes, 27, 1147-1157, 10.1002/hyp.9287,
  2013.
- 711 Lischeid, G.: Non-linear visualization and analysis of large water quality data sets: a model-
- 712 free basis for efficient monitoring and risk assessment, Stochastic Environmental Research

- and Risk Assessment, 23, 977-990, 10.1007/s00477-008-0266-y, 2008.
- Liu, K.-K., Yui, T.-F., Yeh, Y.-H., Tsai, Y.-B., and Teng, T.-L.: Variations of radon content in
- groundwaters and possible correlation with seismic activities in northern Taiwan, pure and
- 716 applied geophysics, 122, 231-244, 1984.
- 717 Manga, M.: Origin of postseismic streamflow changes inferred from baseflow recession and
- 718 magnitude-distance relations, Geophysical Research Letters, 28, 2133-2136,
  719 10.1029/2000gl012481, 2001.
- Manga, M., and Wang, C. Y.: Pressurized oceans and the eruption of liquid water on Europa
  and Enceladus, Geophysical Research Letters, 34, 10.1029/2007gl029297, 2007.
- 722 Manga, M., Beresnev, I., Brodsky, E. E., Elkhoury, J. E., Elsworth, D., Ingebritsen, S. E.,
- Mays, D. C., and Wang, C.-Y.: Changes in permeability caused by transient stresses: Field observations, experiments, and mechanisms, Reviews of Geophysics, 50,
- 725 10.1029/2011rg000382, 2012.
- 726 Matsumoto, N., Kitagawa, G., and Roeloffs, E.: Hydrological response to earthquakes in the
- Haibara well, central Japan–I. Groundwater level changes revealed using state space
  decomposition of atmospheric pressure, rainfall and tidal responses, Geophysical Journal
- 729 International, 155, 885-898, 2003.
- Minissale, A.: Origin, transport and discharge of CO2 in central Italy, Earth-Science Reviews,
  66, 89-141, 10.1016/j.earscirev.2003.09.001, 2004.
- 732 Négrel, P., Giraud, E. P., and Widory, D.: Strontium isotope geochemistry of alluvial
- 733 groundwater: a tracer for groundwater resources characterisation, Hydrology and Earth
- 734 System Sciences Discussions, 8, 959-972, 2004.
- Négrel, P.: Water–granite interaction: clues from strontium, neodymium and rare earth
  elements in soil and waters, Applied Geochemistry, 21, 1432-1454, 2006.
- Noguchi, M., and Wakita, H.: A method for continuous measurement of radon in groundwater
  for earthquake prediction, Journal of Geophysical Research, 82, 1353-1357, 1977.
- Nur, A., and Booker, J. R.: Aftershocks caused by pore fluid flow?, Science, 175, 885-887,1972.
- 741 Park, J. C., Kim, W., Chung, T. W., Baag, C. E., and Ree, J. H.: Focal mechanisms of recent
- earthquakes in the southern Korean Peninsula, Geophysical Journal International, 169, 1103-1114, 2007.
- Park, Y., Ree, J.-H., and Yoo, S.-H.: Fault slip analysis of Quaternary faults in southeastern
- 745 Korea, Gondwana Research, 9, 118-125, 2006.

[revised manuscript text omitted]

Wang, K., Hu, Y., Bevis, M., Kendrick, E., Smalley, R., Vargas, R. B., and Lauría, E.: Crustal

Figure. 1. Study flow that processes observation data and obtains conceptual models.
The used data (groundwater level, temperature, EC, radon-222, and strontium isotopes)
and the adopted method (SOM statistical method) were described with related main
results. At the end of flow map, the conceptual models using the grouping results were
described by the table.

Figure. 2. (a) Location map of the study area and well locations. The upper right map 841 shows the location of Gyeongju area on the southeastern Korean Peninsula. A color 842 scale bar for the elevation map (Elev. in meter) was shown in the right bottom of the 843 figure. Red stars indicate the epicenters of the mainshock, foreshock, and the largest 844 aftershock of the 2016 Gyeongju earthquakes. Magnitudes of the other aftershocks of 845  $M_L \ge 3.0$  are also marked by circles illustrated by the color table. Gyeongju (yellow 846 square) and the well locations (blue squares) are highlighted. (b) Geological map of the 847 study area. The color legend shows the lithostratigraphic units comprising the Gyeongju 848 area. Major faults comprising the Yangsan Fault System are denoted with abbreviations; 849 YSF, Yangsan Fault; MoRF, Moryang Fault, MiRF, Miryang Fault; USF, Ulsan Fault, 850 JNF, Jain Fault. The two maps (Figs. 2(a) and 2(b)) use the same map scale, which is 851 located in the right side of Fig. 2(b). 852

Figure. 3. Time series data of groundwater level in (a) KW 5; (b) KW 8; and (c) KW 11.

The dates of the mainshock and aftershock of the earthquake ( $M_L \ge 4.5$ ) are marked as the orange colored line.

Figure. 4. Time series data of the KW 11 well: (a) temperature and (b) electrical conductivities. The dates of the mainshock and aftershock of the earthquake ( $M_L \ge 4.5$ ) are marked as the orange colored line.

859 **Figure. 5. Spatial distribution of radon concentrations in the Gyeongju area.**

Figure. 6. 87Sr/86Sr vs 1/Sr plot for the groundwater samples. The rectangular boxes indicate each group defined considering the results of both SOM and lithostratigraphy. Green colored box is Group A (shale and sandstone), orange colored box is Group B (granite), yellow colored box is the KW 6 wells of Group C, and the red colored box is Group D (andesite).

Figure. 7. Spatial distribution of strontium concentrations and 87Sr/86Sr ratios in the
Gyeongju area.

Figure. 8. Correlation plot of strontium and calcium values of the groundwater samples
in Gyeongju area.

Figure. 9. Visualization of the component planes of the hydrogeochemical data for the
Gyeongju area from the SOM results. The white indicates the high values of nodes and
the deep brown is the low values of nodes.

41

Figure. 10. U-matrix visualization and pattern of group formation of the SOM results in

the Gyeongju area. The word of a hexagon denotes the sample number.

Figure. 11. (a) Conceptual model to explain the responses of the groundwater system induced by the Gyeongju earthquakes: active water-rock interactions increasing the geochemical signature (KW 9 in Group A), water level anomaly related to non-recoverable deformation (KW 5 in Group B) (dotted line indicates the water table before the earthquakes, the solid line and red inverted triangle indicate the water table after the earthquakes), strong mixing between shallow and deep aquifer caused by sea water intrusion (KW 4 in the Group C), and strike-slip deformation leading to the difference between the alluvial aquifer and the bedrock aquifer (Group D). (b) Simplified geological cross section of KW 6-1.

Fig. S1. Piper diagram with four groups of groundwater samples.

| Date, time           | ML  | Longitude | Latitude |
|----------------------|-----|-----------|----------|
| 2016-11-13, 21:52:57 | 3.5 | 36.36 N   | 126.63 E |
| 2016-11-06, 06:26:22 | 3.5 | 33.76 N   | 125.07 E |
| 2016-09-21, 11:53:54 | 3.5 | 35.75 N   | 129.18 E |
| 2016-09-19, 20:33:58 | 4.5 | 35.74 N   | 129.18 E |
| 2016-09-12, 20:34:22 | 3.6 | 35.78 N   | 129.19 E |
| 2016-09-12, 20:32:54 | 5.8 | 35.76 N   | 129.19 E |
| 2016-09-12, 19:44:32 | 5.1 | 35.77 N   | 129.19 E |

Table 1. The mainshock and aftershocks data (ML  $\geq$  3.5) of the Gyeongju earthquake.

† The bold italics is the mainshock of the Gyeongju earthquakes.

[revised manuscript text omitted]

†KW ##-1 refers the alluvial aquifer well, KW ##-2 or no hyphen well refers the bedrock aquifer well, and KW 11-3 indicates the surface water sample near the KW 11 wells.

| Group | y Well ID    |         | Well type | Groundwater
level | Temperature | EC | Radon con.
( H : > 15000
L : < 800 ) | Radon con.
Difference | Strontium con.
(H:>3000
L:<100) | ${}^{87}Sr/{}^{86}Sr$
( H : > 0.709
L : < 0.707 ) |
|-------|--------------|---------|-----------|----------------------|-------------|----|--------------------------------------------|--------------------------|---------------------------------------|---------------------------------------------------------|
|       | KW 1         |         | Bedrock   |                      |             | 0  |                                            |                          | Н                                     | Н                                                       |
|       | KW2          |         | Bedrock   |                      |             | 0  |                                            |                          | Н                                     | Н                                                       |
|       | KW O  | KW 9-1  | Alluvial  |                      |             |    | L                                          |                          |                                       |                                                         |
| A     | NW 9  | KW 9-2  | Bedrock   |                      |             |    | L                                          |                          |                                       |                                                         |
|       | WW 10 | KW 10-1 | Alluvial  |                      |             |    | L                                          |                          |                                       |                                                         |
|       | KW 10        | KW 10-2 | Bedrock   |                      |             | 0  | L                                          |                          | Н                                     | Н                                                       |
| В     | K            | CW 3    | Bedrock   |                      |             |    |                                            |                          | L                                     | L                                                       |
|       | VW 5         | KW 5-1  | Alluvial  | 0                    |             |    | Н                                          |                          |                                       |                                                         |
| В     | KW D         | KW 5-2  | Bedrock   | 0                    |             |    | Н                                          |                          |                                       |                                                         |
|       | KW 12        | KW 12-1 | Alluvial  |                      |             |    |                                            |                          |                                       |                                                         |
|       |              | KW 12-2 | Bedrock   |                      |             |    |                                            |                          |                                       |                                                         |
|       | KW 4         | KW 4-1  | Alluvial  |                      |             |    |                                            | T                        | L                                     |                                                         |
|       |              | KW 4-2  | Bedrock   |                      |             |    |                                            | L                        |                                       |                                                         |
| C     | NM C  | KW 6-1  | Alluvial  |                      |             | 0  | L                                          | т                        | L                                     | L                                                       |
| C     | KWO          | KW 6-2  | Bedrock   |                      |             | 0  | L                                          | L                        | L                                     | Н                                                       |
|       | VW 7         | KW 7-1  | Alluvial  |                      |             |    |                                            | т                        |                                       | L                                                       |
|       | KW/          | KW 7-2  | Bedrock   |                      |             |    | L                                          | L                        | L                                     |                                                         |
|       | VW 9         | KW 8-1  | Alluvial  | 0                    |             |    |                                            | TT                       |                                       |                                                         |
| D     | KW 8         | KW 8-2  | Bedrock   | 0                    |             |    | Н                                          | П                        | L                                     | L                                                       |
| D     | WW 11 | KW 11-1 | Alluvial  | 0                    | 0           | 0  | L                                          | ŢŢ                       | L                                     | L                                                       |
|       | KW II        | KW 11-2 | Bedrock   | 0                    | 0           | 0  |                                            | H                        |                                       | Н                                                       |

Table 4. Anomaly data of groundwater wells based on the grouping results.

† O' refers that the anomaly was detected, 'H' refers the high concentration, and 'L' refers the low concentration.

---

## Author Comment (AC1)

*Dear Editor and Referee,*

*Taking into account the useful comments we received until now, we provided a new version of the manuscript with adequate corrections. We enclose below our responses to all Referee's comments, which surely improve the initial manuscript. For the moment we do not include the new version of the manuscript but we are ready to submit when required. Otherwise, the totally revised manuscript will be uploaded after reflecting other reviewer's comments. We are confident that provided changes are sufficient for reconsidering our manuscript, and we wait for additional comments by the same or other reviewers.*

*Best Regards,*

*For the Authors*

**Comments provided by Anonymous Referee #1**

**1) general comments**

The manuscript is a technical report which provides an interesting analysis of hydrological data and discusses the evolution of hydrological quantities before, during and after some earthquakes in South Korea.

The work is largely based on well-established methods and the scientific novelty is not very remarkable. The main goal is stated in the introduction at lines 105 to 107 and then later at lines 357 & 358. However, the neural network method applied in this paper (SOM - Self-Organizing Map) is not new and it is not apparent its advantage with respect to other clustering methods. See also the specific comment # 4.

The manuscript is generally well organized and written, but it requires linguistic improvements, some of which are listed in the technical comments below. From the scientific point of view, at some points, the manuscript is not sufficiently precise and rigorous, as described, e.g., in the specific comments # 1, 2 & 3 and in the technical comments # 9 & 10 below.

I am sorry, but I think that the innovative character of the work is not sufficient to publish the paper on HESS, whereas the manuscript is more adequate for a strictly hydrogeological journal.

RESPONSE:

We thank the reviewer for taking his/her time to review our paper. We have attempted to satisfy all suggestions, so it made us to produce a stronger paper, adequate to be published on HESS. Please see the responses to the reviewers' comments below and subsequent changes in the revised manuscript (marked in red color). Thanks in advance for reconsidering the revised manuscript positively.

The novelty of this paper is to combine the hydrologic, hydrogeochemical, and lithostratigraphic characteristics by applying the neural network method 'SOM'. The SOM method has an advantage in visualization of the multi-dimensional data, which is helpful to identify the dependencies between the variables (e.g. hydrogeochemical and isotopic data) and to classify the wells. Especially, it provides the detailed local relationship between the variables by the component planes. The local interpretation is important for the studies related to the earthquakes. The clustering analysis of the SOM is not special compared to other statistical methods, however, this paper shows the interesting results that the grouping was in accordance with the lithostratigraphic unit. This is powerful results to understand the hydrologic response to the earthquakes considering the geologic characteristics, which can be applied to other sites. Moreover, there are few cases using SOM for the groundwater study related to the earthquakes, so we suggest the application of SOM to researches in other sites for making statistically explanatory basis and then provide geological and hydrogeological interpretations of the observed phenomena.

The manuscript was checked by a professional editing service. After revision work of other reviewers, we will check final manuscript one more time by other editing service.

**2) Specific comments**

1. Lines 37 to 41. The list of references shows that it is debatable to state that "few" studies

are devoted to this research topic. A simple and fast search on google scholar shows a lot of papers related to the effects of earthquakes on hydrological processes and quantities. Perhaps, the Authors want to stress that most papers are devoted to earthquake precursors or to the study of co-seismic phenomena.

RESPONSE:

We agree that the statement might cause misunderstanding. According to your comments, we have rewritten those sentences.

[L35-38] Typically, most studies have focused on earthquake forecasting, i.e. changes prior to earthquakes, or co-seismic behavior. There have been a limited number of studies that discuss the responses of groundwater systems, especially, emphasize the hydrogeologic changes after earthquakes.

2. Throughout the paper, it would be necessary to consider in a more accurate and rigorous way the considered time scales. The following remarks provide some instances.

• Lines 43 & 44. If the changes are related to seismic waves, they should disappear after the earthquake. Effects at different time-scales should be separated more clearly. The sentence "Seismic waves... geochemistry" should be better connected with the preceding one "Seismicity... groundwater systems".

• Line 187. The time scales should be considered in a more accurate and rigorous way. In fact, the sampling period of hydrological data is high with respect to the duration of the earthquake wave train.

RESPONSE:

We have moved "Seismic waves... geochemistry" sentence to follow "Seismicity... groundwater systems" sentence (L41-45).

In line 187, we explained the information with erroneous descriptions, so we corrected that sentence. [L185-187] The hourly data, which were monitored every hour on the hour, were used to observe the responses before, during, and after the earthquake. [L188-189] These daily data correspond to the cumulative quantities during the day.

3. Line 93. Surface area is an extensive property: does radon concentration (which is an intensive property) depend on it? Should "surface area" be substituted with the intensive property "specific surface"?

RESPONSE:

We agree with your comment. "Surface area" was replaced by "specific surface" (L88-89).

4. In section 5 it is shown that several different processes might explain the behavior of collected data. However, most (if not all) of such processes have been hypothesized in previously published papers: many of the papers cited in this section were published in the second half of the XX century. Such a discussion has a great value for local land management and natural risks mitigation, but a more limited interest for the international scientific community. Moreover, the declared goal of the paper is to show the relevance of the use of SOM, but the discussion of the relevance of this method – as compared with other possible approaches – is almost absent.

RESPONSE:

We have added some references recently published as:

[L337-338] (Fleeger et al., 1999; Kitagawa et al., 2006; Rojstaczer and Wolf, 1992; Rojstaczer et al., 1995; Wang et al., 2004)

[L349] (Claesson et al., 2007; Hartmann and Levy, 2005)

[L387] (Sparks, 2003)

[L427] (Bullen et al., 1997; Franklyn et al., 1991; Négrel, 2006)

[L442-443] (King and Cocco, 2001; Nur and Booker, 1972; Peng and Zhao, 2009; Scholz, 2002; Scholz et al., 1973)

Moreover, we have split the discussion part into three chapters for more clearly and logically interpretation as;

5.1 Groundwater level, temperature, and EC changes

5.2 Isotopic data (radon and strontium)

5.3 The conceptual model with the SOM method

The additional explanations including 'see Fig. 3' mark were written in discussion part for showing the relevance of the use of SOM with the conceptual model.

[L372-375] The dependencies between the variables, hydrochemical parameters including strontium isotopes, could be interpreted with the component plane results from the SOM (see Fig. 3). These correlations were also used for analyzing the possible mechanisms at each group.

[L379] low $^{87}Sr/^{86}Sr$ ratio and low pH (see Fig. 3 and Fig. 10)

[L385-389] The capacity of the cation-bearing soil (cation-exchange capacity; CEC) depends on the pH of the surrounding water, and the CEC of a soil generally show decreases with pH decreases (Sparks, 2003). The acidic water of KW10-2 (pH = 2.27) would lead to lower CEC and lead to leaching of $Ca^{2+}$ and $Sr^{2+}$ from the soil to surrounding groundwater.

[L411-412] Moreover, KW 4-1, KW 4-2, and KW 6-2 had high values of EC, Cl, TDS, and salinity values (see Fig.3)

[L414-415] The strontium concentration and Ca values are also low in these wells (see Fig. 3 and Fig. 10).

[L426-427] The former occurs more rapidly, providing $Ca^{2+}$ and $Sr^{2+}$ with a low $^{87}Sr/^{86}Sr$ ratio (see Fig. 3) (Bullen et al., 1997; Franklyn et al., 1991; Négrel, 2006).

5. Carefully revise the number of significant digits used for several quantities. For instance, with reference to Table 3, is it physically significant to express concentrations with three or four significant digits, radon activity with up to five significant digits and Sr isotopic ratio with six significant digits?

RESPONSE:

We have corrected the significant digits totally.

In general, the ion concentrations used two-four significant digits and $^{87}Sr/^{86}Sr$ used six-eight significant digits. The significant digits of ion concentration in wrong place were united in accordance with the digits. Please check this work in Table 3. Moreover, the significant digits in the text were also adjusted to the indicated digits as shown in Table 3.

The radon concentration used two or three significant digits in pCi/L, three or four significant digits in Bq/L, and three-six significant digits in Bq/m$^3$. The radon concentration in groundwater was mostly expressed by the unit of in Bq/L or Bq/m$^3$. The measurement device, RTM1688-2 of SARAD, expressed the radon concentration in water as the unit 'Bq/m$^3$'. After measurement, the radon concentrations were calibrated by the decay equation considering the sampling time and measurement time, so we used these values without changing significant digits.

[L299-302] The $^{87}Sr/^{86}Sr$ values ranged from 0.705688 to 0.712368 (see Table 3). In the alluvial aquifer wells, the $^{87}Sr/^{86}Sr$ values ranged from 0.706191 to 0.708353, and these values were from 0.705688 to 0.712368 in the bedrock aquifer wells.

[L305-306] The $^{87}Sr/^{86}Sr$ values of the Bulguksa group (KW 3, KW 5-2, and KW 12-2) ranged from 0.706575 to 0.708022.

[L356-358] Generally, Cretaceous granites comprising the Gyeongsang basin had a strontium concentration from 62 ppm to 428 ppm and an $^{87}Sr/^{86}Sr$ ratio from 0.704610 to 0.711400 (Cheong and Jo, 2017).

[L358-360] Basaltic rocks near the Yeonil group and Janggi group had strontium concentration from 439 ppm to 518 ppm and an $^{87}Sr/^{86}Sr$ ratio from 0.703850 to 0.704630 (Shimazu et al., 1990).

[L360-362] In the Chaeyaksan basaltic volcanics of the Yucheon group, strontium concentration ranged from 731 ppm to 1667 ppm and the $^{87}Sr/^{86}Sr$ ratio from 0.705870 to 0.706440 (Yun, 1998).

3) Technical comments

1. Line 9. Specify $M_L$.

RESPONSE:

We have corrected it as:

[L9-11] The September 12, 2016 Gyeongju earthquakes ($M_L = 5.1$ of foreshock and $M_L = 5.8$ of mainshock) had significant effects on groundwater systems along the Yangsan Fault System composed of NNE-trending, right-lateral strike-slip faults in Korea.

2. Lines 15 & 16. Rephrase "with bedrock characteristics".

RESPONSE:

We have changed that from "with bedrock characteristics" to "with lithostratigraphic classification" (L16).

3. Lines 16 to 19. Erase "To analyze... from the earthquakes" and rephrase the remaining part of the sentence "annual monitoring data... during January 2017".

RESPONSE:

We have rewritten this sentence as:

[L16-18] For this, annual monitoring data (groundwater level, temperature, and electrical conductivity) and collected data (hydrochemical parameters, radon-222, and strontium isotopes) were used.

4. Line 35. Substitute "underground" with "groundwater".

RESPONSE:

It was done. We have replaced "underground" word to "groundwater" (L34).

5. Line 55. Rephrase "By using hydraulic properties".

RESPONSE:

We have rewritten that sentence as:

[L55-56] Some studies also have proposed some conceptual models for describing the aquifer responses to earthquakes by analyzing hydraulic properties.

6. Line 61. Rephrase "was recorded as the largest".

RESPONSE:

We have rephrased the sentence as:

[L61-62] The mainshock of the Gyeongju events was recorded as the largest earthquake in Korea since instrumental seismic monitoring started in Korea in 1978.

7. Line 64 to 69. The sentences "The occurrence... near the YSF (Lee and Jin, 1991;Lee and Na, 1983)" could be erased, since they do not give any scientific information relevant for the paper's objectives.

RESPONSE:

Thank you for your comment to make the paragraph contextually much better. We have erased the sentence completely.

8. Line 69. Substitute "interpreted" with "shown".

RESPONSE:

We have corrected the vocabulary; from "interpreted" to "shown" (L65).

9. Lines 72 to 75. The sentence "The occurrence... following the Gyeongju earthquakes" is

quite self-evident and could be better rephrased.

RESPONSE:

We have rewritten the sentence as:

[L67-70] Gyeongju area is spatially close to the YSF, thus the Gyeongju events might allow more detailed studies about characteristics of the YSF and its branch faults, including groundwater responses after earthquakes.

10. Lines 81 & 82. Please explain how temperatures can be derived from measurements of groundwater level.

RESPONSE:

We have rewritten the sentence as:

[L76-77] Temperature changes are commonly analyzed using heat transport modeling (EkemenKeskin, 2010; Wang et al., 2012).

11. Lines 97 & 98. Add references.

RESPONSE:

We have added references (L93-94).

1.Adinolfi Falcone, R., Carucci, V., Falgiani, A., Manetta, M., Parisse, B., Petitta, M., Rusi, S., Spizzico, M., and Tallini, M.: Changes on groundwater flow and hydrochemistry of the Gran Sasso carbonate aquifer after 2009 L'Aquila earthquake, Italian Journal of Geosciences, 131, 459-474, 2012.
2. Igarashi, G., and Wakita, H.: Groundwater radon anomalies associated with earthquakes, Tectonophysics, 180, 237-254, 1990.
3. Igarashi, G., Tohjima, Y., and Wakita, H.: Time-variable response characteristics of groundwater radon to earthquakes, Geophysical research letters, 20, 1807-1810, 1993.

12. Line 99. Specify "these": strontium only or radon and strontium?

RESPONSE:

We have specified from "these" to "Strontium isotopes" (L95).

13. Line 101. Rephrase "according to the rock type in the bedrock of aquifers"

RESPONSE:

We hope that the sentence below may be more clearly than the previous one.

[L97-98] The $^{87}Sr/^{86}Sr$ ratios of the bedrock aquifers were different according to rock types of bedrock.

14. Line 129. Substitute "is approximately 21–35 km" with "varies between 21 km and 35 km".

RESPONSE:

We have changed the phrase "is approximately 21–35 km" to "varies between 21 km and 35 km" (L125).

15. Lines 235 to 237. Rephrase, possibly as "The raw data were normalized in order to work with transformed quantities with zero mean and unit standard deviation".

RESPONSE:

Thank you for your suggestion. We have revised it according to your suggestion (L240-241).

16. Line 238. Substitute "have" with "show".

RESPONSE:

We have replaced from "have" to "show" (L241).

17. Line 287. Correct the exponent in Bq $\cdot$ m$^{-3}$.

RESPONSE:

We have corrected that.

18. Line 333 & 334. Rephrase the sentence "Groundwater level oscillation... of the aquifer".

RESPONSE:

We have rephrased the sentence as:

[L339-341] Groundwater level oscillation also depends on the interactions between inflow/outflow of the well and of the aquifer (Cooper et al., 1965).

19. Line 355. Rephrase "were expected compared to those".

RESPONSE:

We have rewritten the whole sentence to make the meaning clear, in addition to rephrasing the previous "were expected compared to those".

[L362-364] Thus, samples of the Bulguksa granite would be more radiogenic than those of the Yucheon group rocks, because of the composite minerals of the Bulguksa granite (plagioclase, feldspar, and biotite) which have high $^{87}$Sr/$^{86}$Sr ratios.

20. Line 359. Rephrase "which was useful".

RESPONSE:

We have rephrased to "which was used" and deleted ", based on bedrock characteristics" (L368).

21. Line 363. Rephrase "as one well binding the alluvial and bedrock aquifer wells".

RESPONSE:

We have deleted the "as one well" which is unnecessary description (L372).

22. Line 382. Substitute "885–7851 ppb" with "from 885 ppb to 7851 ppb".

RESPONSE:

We agree and made changes (L393-394).

23. Line 476. What about hydrogeochemical data?

RESPONSE:

We have added the sentence "The hydrogeochemical dataset was shown in Table 3." (L489).

24. Line 765. Use capital letters for "C.-Y.".

RESPONSE:

We have corrected that.

25. Table 3. Rephrase the text of the footnote. Moreover, information is given for KW *-1 and KW *-2: what about KW 11-3? I am afraid that further details are missing: are the screened intervals located at different depths, or are these clusters of wells with different depths?

RESPONSE:

We have written additional information about that.

[Table 3] [†]KW ##-1 refers the alluvial aquifer well, KW ##-2 or no hyphen well refers the bedrock aquifer well, and KW 11-3 indicates the surface water sample near the KW 11 wells.

The further details of the wells were also added in the text.

[L152-156] The wells generally were installed by two types at each point. The one well (as labeled KW ##) indicates that the sampling point is consisted of only one type well, bedrock aquifer well. The alluvial aquifer wells were labeled KW##-1 and the bedrock aquifer wells were labeled KW## or KW##-2. The KW 11-3 refers the surface water sample near the KW 11 well.

26. Figure 1. Sorry, but I do not understand this Figure. A more accurate and thorough description in the figure caption is necessary.

RESPONSE:

We have redrawn the figure 1 and added further information in the caption.

[L819-823] Study flow that processes observation data and obtains conceptual models. The used data (groundwater level, temperature, EC, radon-222, and strontium isotopes) and the adopted method (SOM statistical method) were described with related main results. At the end of flow map, the conceptual models using the grouping results were described by the table.

27. Figure 2. The colour scale of map (a) is a representation of ground surface level, isn't it? Maps (a) and (b) have the same extension, haven't they? Add this information in the figure caption and the colour scale bar in map (a).

RESPONSE:

We are sorry for our carelessness. Colorbar for the elevation (Elev. in meter) is now added to the right side of the bottom of Fig. 2. The information about the colorbar and the map scale is also added to the figure caption of Fig.2.

[L825-827] A color scale bar for the elevation map (Elev. in meter) was shown in the right bottom of the figure.

[L834-835] The two maps (Figs. 2(a) and 2(b)) use the same map scale, which is located in

the right side of Fig. 2(b).

28. Figures 3 & 4. These representations are not easily interpreted. More details in the text and in the figure captions could be useful.

RESPONSE:

We have added more interpretation of the component plane as:

[L234-238] This method also provides the detailed local relationship between the variables by the component planes, which is helpful to understand groundwater systems visually. The contribution map of the variables is shown in the component map (Fig. 3). Each component plane represents the average component value at each node in a certain color; the white indicates the high values and the deep brown indicates the low values.

[L249-250] Deep brown shades on the U-matrix indicate a large distance between neighborhood nodes whereas white shades correspond to a short distance between nodes.

[L837-838]The white indicates the high values of nodes and the deep brown is the low values of nodes.

[L840] The word of a hexagon denotes the sample number.